# VideoJAM: Joint Appearance-Motion Representations for Enhanced Motion Generation in Video Models

**Hila Chefer** [*1 2]   **Uriel Singer** [1]   **Amit Zohar** [1]   **Yuval Kirstain** [1]
**Adam Polyak** [1]   **Yaniv Taigman** [1]   **Lior Wolf** [2]   **Shelly Sheynin** [1]
https://hila-chefer.github.io/videojam-paper.github.io/

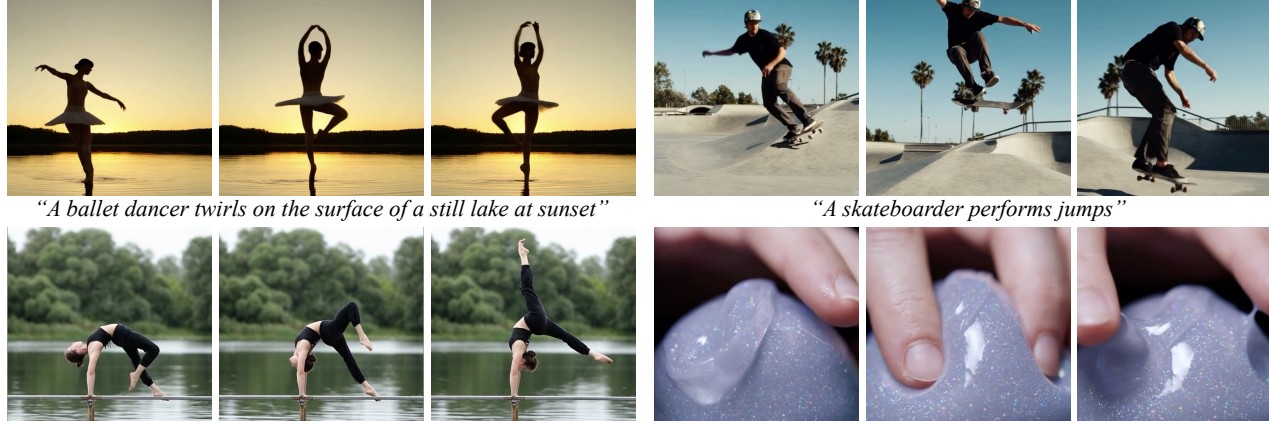

*"A ballet dancer twirls on the surface of a still lake at sunset"*

*"A skateboarder performs jumps"*

*"An acrobat executing a handstand on a narrow beam above water"*

*"Fingers press into a shimmering slime ball"*

*Figure 1.* **Text-to-video samples generated by VideoJAM.** We present VideoJAM, a framework that explicitly instills a strong motion prior to any video generation model. Our framework significantly enhances motion coherence across a wide variety of motion types.

## Abstract

Despite tremendous recent progress, generative video models still struggle to capture real-world motion, dynamics, and physics. We show that this limitation arises from the conventional pixel reconstruction objective, which biases models toward appearance fidelity at the expense of motion coherence. To address this, we introduce **Video-JAM**, a novel framework that instills an effective motion prior to video generators, by encouraging the model to learn *a joint appearance-motion representation*. VideoJAM is composed of two complementary units. During training, we extend the objective to predict both the generated pixels and their corresponding motion from a single learned representation. During inference, we introduce **Inner-Guidance**, a mechanism that steers the generation toward coherent motion by lever-aging the model's own evolving motion prediction as a dynamic guidance signal. Notably, our framework can be applied to any video model with minimal adaptations, requiring no modifications to the training data or scaling of the model. VideoJAM achieves state-of-the-art performance in motion coherence, surpassing highly competitive proprietary models while also enhancing the perceived visual quality of the generations. These findings emphasize that appearance and motion can be complementary and, when effectively integrated, enhance both the visual quality and the coherence of video generation.

## 1. Introduction

Recent advances in video generation showcased remarkable progress in producing high-quality clips (Brooks et al., 2024; KlingAI, 2024; Polyak et al., 2024). Yet, despite continuous improvements in the visual quality of the generated videos, these models often fail to accurately portray motion, physics, and dynamic interactions (Kang et al., 2024; Brooks et al., 2024) (Fig. 2). When tasked with generating challenging motions like gymnastic elements (e.g., a cartwheel in Fig. 2(b)), the generations often display severe

---
[*]Work done while the first author was an intern at GenAI, Meta. [1]GenAI, Meta [2]Tel Aviv University. Correspondence to: Hila Chefer <hilach70@gmail.com>.

*Proceedings of the 42nd International Conference on Machine Learning*, Vancouver, Canada. PMLR 267, 2025. Copyright 2025 by the author(s).

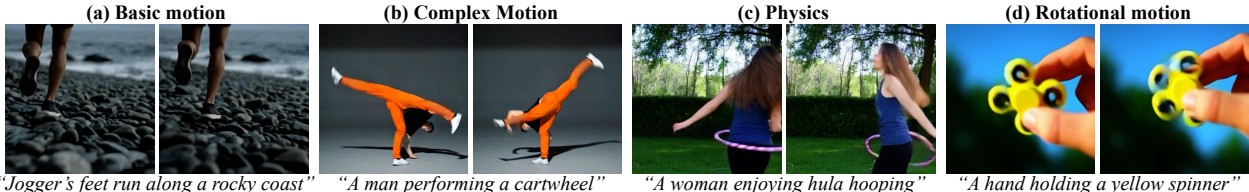

**(a) Basic motion**     **(b) Complex Motion**     **(c) Physics**     **(d) Rotational motion**

*"Jogger's feet run along a rocky coast"*    *"A man performing a cartwheel"*    *"A woman enjoying hula hooping"*    *"A hand holding a yellow spinner"*

*Figure 2.* **Motion incoherence in video generation.** Examples of incoherent generations by DiT-30B (Peebles & Xie, 2023). The model struggles with (a) basic motion, e.g., jogging (stepping on the same leg repeatedly); (b) complex motion e.g., gymnastics; (c) physics, e.g., object dynamics (the hoop passes through the woman); and (d) rotational motion, failing to replicate simple repetitive patterns.

deformations, such as the appearance of additional limbs. In other cases, the generations exhibit behavior that contradicts fundamental physics, such as objects passing through other solid objects (e.g., a hula hoop passing through a woman in Fig. 2(c)). Another example is rotational motion, where models struggle to replicate a simple repetitive pattern of movement (e.g., a spinner in Fig. 2(d)). Interestingly, these issues are prominent even for basic motion types that are well-represented in the model's training data (e.g., jogging in Fig. 2(a)), suggesting that data and scale may not be the sole factors responsible for temporal issues in video models.

In this work, we aim to provide insights into why video models struggle with temporal coherence and introduce a generic solution that achieves state-of-the-art motion generation results. First, we find that the gap between pixel quality and motion modeling can be largely attributed to the common training objective. Through qualitative and quantitative experiments (see Sec. 3), we show that the pixel-based objective is *nearly invariant to temporal perturbations* in generation steps that are critical to determining motion.

Motivated by these insights, we propose **VideoJAM**, a novel framework that equips video models with an explicit motion prior by teaching them a **J**oint **A**ppearance-**M**otion representation. This is achieved through two complementary modifications: during training, we amend the objective to predict motion in addition to appearance, and during inference, we propose a guidance mechanism to leverage the learned motion prior for temporally coherent generations.

Specifically, during the VideoJAM training, we pair the videos with their corresponding motion representations and modify the network to predict both signals (appearance and motion). To accommodate this dual format, we only add two linear layers to the architecture (see Fig. 4). The first, located at the input to the model, combines the two signals into a single representation. The second, at the model's output, extracts a motion prediction from the learned joint representation. The objective function is then modified to predict the joint appearance-motion distribution, encouraging the model to rely on the added motion signal.

At inference, our primary objective is video generation, with the predicted motion serving as an auxiliary signal. To guide the generation to effectively incorporate the learned motion

prior, we introduce **Inner-Guidance**, a novel inference-time guidance mechanism. Unlike existing approaches (Ho & Salimans, 2022; Brooks et al., 2023), which depend on fixed external signals, Inner-Guidance leverages the model's own evolving motion prediction as a dynamic guidance signal. This setting requires addressing unique challenges: the motion signal is inherently dependent on the other conditions and the model weights, making the assumptions of prior works invalid and requiring a new formulation (Sec. 2, App. A). Our mechanism directly modifies the model's sampling distribution to steer the generation toward the joint appearance-motion distribution and away from the appearance-only prediction, allowing the model to refine its own outputs throughout the generation process.

Through extensive experiments, we demonstrate that applying VideoJAM to pre-trained video models significantly enhances motion coherence across various model sizes and diverse motion types. Furthermore, VideoJAM establishes a new state-of-the-art in motion modeling, surpassing even highly competitive proprietary models. These advances are achieved without the need for any modifications to the data or model scaling. With an intuitive design requiring only the addition of two linear layers, VideoJAM is both generic and easily adaptable to any video model. Interestingly, Video-JAM also improves the perceived quality of the generations, even though we do not explicitly target pixel quality. These findings underscore that appearance and motion are not mutually exclusive but rather inherently complementary.

## 2. Related Work

Diffusion models (Ho et al., 2020) revolutionized visual content generation. Beginning with image generation (Dhariwal & Nichol, 2021; Rombach et al., 2022; Ho et al., 2022a; Black Forest Labs, 2024; Dai et al., 2023; OpenAI, 2024), editing and personalization (Gal et al., 2022; Ruiz et al., 2023; Chefer et al., 2024b;a), and more recently video generation. The first efforts to employ diffusion models for videos relied on model cascades (Ho et al., 2022b; Singer et al., 2023) or direct "inflation" of image models using temporal layers (Guo et al., 2023; BarTal et al., 2024; Wu et al., 2023). Other works focused on adding an auto-encoder for efficiency (Blattmann et al., 2023b; An et al., 2023;

Wang et al., 2023), or conditioning the generation on images (Blattmann et al., 2023a; Girdhar et al., 2024; Hong et al., 2022). Recently, the UNet backbone was replaced by a Transformer (Polyak et al., 2024; Brooks et al., 2024; Genmo, 2024; HaCohen et al., 2024), mostly following Diffusion Transformers (DiTs) (Peebles & Xie, 2023).

To control the generated content, Dhariwal & Nichol (2021) introduced *Classifier Guidance*, where classifier gradients guide the generation toward a specific class. Ho & Salimans (2022) proposed *Classifier-Free Guidance (CFG)*, replacing classifiers with text. Similar to Inner-Guidance, CFG modifies the sampling distribution. However, CFG does not address noisy conditions or multiple conditions. Closest to our work, Liu et al. (2022), handle multiple conditions, $c_1, \ldots, c_n$, using a compositional score estimate,

$$p_\theta(x|c_1, \ldots, c_n) = \frac{p_\theta(x, c_1, \ldots, c_n)}{p_\theta(c_1, \ldots, c_n)}$$

$$\propto p_\theta(x, c_1, \ldots, c_n) = p_\theta(x) \prod_{i=1}^{n} p_\theta(c_i|x).$$

where $\theta$ denotes the model weights and $p$ is the sampling distribution. The above assumes that $c_1, \ldots, c_n$ are independent of each other and $\theta$, which does not hold in our case, since the motion is directly predicted by the model and thus inherently depends on $\theta$ and the conditions. Similarly, Brooks et al. (2023) assume independence between the conditions and model weights $\theta$, which is, again, incorrect in our setting. See App. A for further discussion.

The gap between pixel quality and temporal coherence is a prominent issue (Ruan et al., 2024; Brooks et al., 2024; Liu et al., 2024b; Kang et al., 2024). Previous works explored motion or physics-based signals to improve video generation. Some methods use them as *input* for guidance or editing (Geng et al., 2024; Ma et al., 2023; Liu et al., 2024a; Cong et al., 2023; Montanaro et al., 2024). Note that their objective differs from ours since we aim to *teach* models a temporal prior rather than taking it as input. Other methods suggest separating content and motion generation (Tulyakov et al., 2018; Ruan et al., 2024; Qing et al., 2023; Shen et al., 2024; Jin et al., 2024b). In the context of image generation, Liu et al. (2023) improves the realism of human images by incorporating spatial priors such as surface-normal and depth maps in addition to a conditioning skeleton given as input. Finally, most similar to our approach, recent works use motion representations to improve coherence in image-to-video generation (Shi et al., 2024; Wang et al., 2024), but these are limited to models conditioned on images.

## 3. Motivation

During training, generative video models take a noised training video and compute a loss by comparing the model's

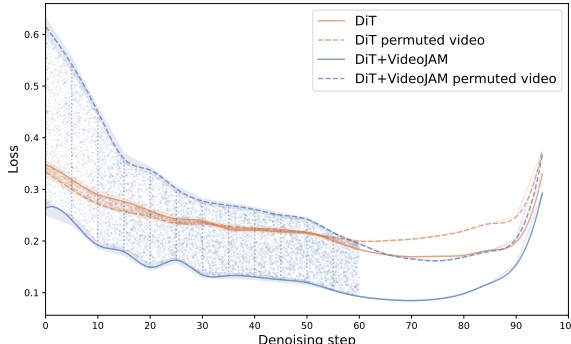

*Figure 3.* **Motivation Experiment.** We compare the model's loss before and after randomly permuting the video frames, using a "vanilla" DiT (orange) and our fine-tuned model (blue). The original model is *nearly invariant* to temporal perturbations for $t \leq 60$.

prediction with the original video, the noise, or a combination of the two (Ho et al., 2020; Lipman et al., 2023) (Sec. 4.1). We hypothesize that this formulation biases the model towards appearance-based features, such as color and texture, as these dominate pixel-wise differences. Consequently, the model is less inclined to attend to temporal information, such as dynamics or physics, which contribute less to the objective. To demonstrate this claim, we perform experiments to evaluate the sensitivity of the model to temporal incoherence. The following experiments are conducted on DiT-4B (Peebles & Xie, 2023) for efficiency.

We conduct an experiment where two variants of videos are noised and fed to the model—first, the plain video without intervention, and second, the video after applying a *random permutation* to its frames. Assuming the model captures temporal information, we anticipate that the temporally incoherent (perturbed) input will result in a higher measured loss compared to the temporally coherent input.

Given a random set of $35,000$ training videos, we noise each video to a random denoising step $t \in [0, 99]$. We then examine the difference in the loss measured before and after the permutation and aggregate the results per timestep. We consider two models– the "vanilla" DiT, which employs a pixel-based objective, and our fine-tuned VideoJAM model, which adds an explicit motion objective (Sec. 4).

The results of this experiment are reported in Fig. 3. As can be observed, the original model appears to be *nearly invariant* to frame shuffling until step $60$ of the generation. This implies that the model fails to distinguish between a valid video and a temporally incoherent one. In stark contrast, our model is extremely sensitive to these perturbations, as is indicated by the significant gap in the calculated loss.

In App. B we include a qualitative experiment demonstrating that the steps $t \leq 60$ determine the coarse motion in the video. Both results suggest that the training objective is less sensitive to temporal incoherence, leading models to favor appearance over motion.

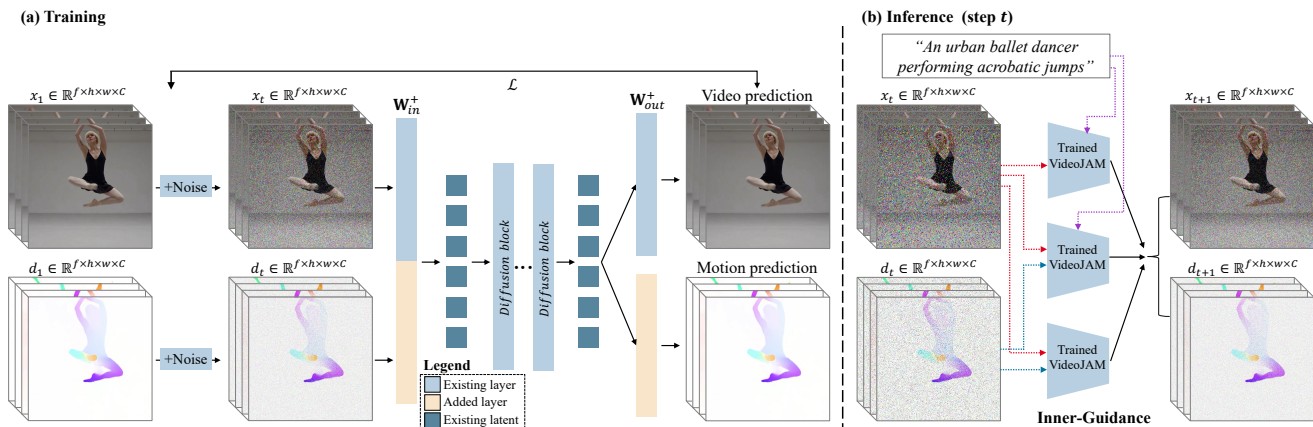

*Figure 4.* **VideoJAM Framework.** VideoJAM is constructed of two units; (a) **Training.** Given an input video $x_1$ and its motion representation $d_1$, both signals are noised and embedded to a *single, joint* latent representation using a linear layer, $\mathbf{W}_{in}^+$. The diffusion model processes the input, and two linear projection layers predict both appearance and motion from the joint representation. (b) **Inference.** We propose *Inner-Guidance*, where the model's own noisy motion prediction is used to guide the video prediction at each step.

## 4. VideoJAM

Motivated by the insights from the previous section, we propose to teach the model a joint representation encapsulating both appearance and motion. Our method consists of two complementary phases (see Fig. 4): (i) During training, we modify the objective to predict the joint appearance-motion distribution; This is achieved by altering the architecture to support a dual input-output format, where the model predicts both the appearance and the motion of the video. (ii) At inference, we add Inner-Guidance, a novel formulation that employs the predicted motion to guide the generated video toward coherent motion.

### 4.1. Preliminaries

We conduct our experiments on the Diffusion Transformer (DiT) architecture, which has become the standard backbone for video generation (Brooks et al., 2024; Genmo, 2024). The model operates in the latent space of a Temporal Auto-Encoder (TAE), which downsamples videos spatially and temporally for efficiency. We use Flow Matching (Lipman et al., 2023) to define the objective. During training, given a video $x_1$, random noise $x_0 \sim \mathcal{N}(0, I)$, and a timestep $t \in [0, 1]$, $x_1$ is noised using $x_0$ to obtain an intermediate latent as follows,

$$x_t = tx_1 + (1 - t) x_0. \quad (1)$$

The model is then optimized to predict the velocity, namely,

$$v_t = \frac{dx_t}{dt} = x_1 - x_0. \quad (2)$$

Thus, the objective function employed for training becomes,

$$\mathcal{L} = \mathbb{E}_{x_1, x_0 \sim \mathcal{N}(0,1), y, t \in [0,1]} \left[ ||u(x_t, y, t; \theta) - v_t||_2^2 \right], \quad (3)$$

where $y$ is an (optional) input condition, $\theta$ denotes the weights, and $u(x_t, y, t; \theta)$ is the prediction by the model.

The prediction, $u$, is obtained using the DiT. First, the model "patchifies" $x_t$ into a sequence of $p \times p$ video patches. This sequence is projected into the DiT's embedding space via a linear projection, $\mathbf{W}_{in} \in \mathbb{R}^{p^2 \cdot C_{\text{TAE}} \times C_{\text{DiT}}}$, where $C_{\text{TAE}}$ and $C_{\text{DiT}}$ are the embedding dimensions of the TAE and DiT, respectively. The DiT then applies stacked attention layers to produce a latent representation for the video, which is projected back to the TAE's space to yield the final prediction using $\mathbf{W}_{out} \in \mathbb{R}^{C_{\text{DiT}} \times C_{\text{TAE}} \cdot p^2}$, i.e.,

$$u(x_t, y, t; \theta) = \mathcal{M}(x_t \cdot \mathbf{W}_{in}, y, t; \theta) \cdot \mathbf{W}_{out}, \quad (4)$$

where $\mathcal{M}$ denotes the attention blocks. For efficiency, we employ models that are pre-trained as described above and fine-tune them using VideoJAM as explained next.

### 4.2. Joint Appearance-Motion Representations

We begin by describing the motion representation employed by VideoJAM. We opt to use optical flow since it is flexible, generic, and easily represented as an RGB video; thus, it does not require training an additional TAE. Optical flow computes a dense displacement field between pairs of frames. Given two frames $I_1, I_2 \in \mathbb{R}^{H \times W \times 3}$, the optical flow, $d \in \mathbb{R}^{H \times W \times 2}$, holds that $d(u, v)$ is the displacement of the pixel $(u, v)$ from $I_1$ in $I_2$. To convert $d$ into an RGB image, we compute the angle and norm of each pixel,

$$m = \min \left\{ 1, \frac{\sqrt{u^2 + v^2}}{\sigma \sqrt{H^2 + W^2}} \right\}, \alpha = \arctan 2(v, u), \quad (5)$$

where $m$ is the normalized motion magnitude, $\sigma = 0.15$, and $\alpha$ is the motion direction (angle). Each angle is assigned a color and the pixel opacity is determined by $m$. Our normalization enables the model to capture motion magnitude, with larger movements corresponding to higher $m$ values

and reduced opacity. By using a coefficient $\sigma = 0.15$ instead of the full resolution ($\sqrt{H^2 + W^2}$), we prevent subtler movements from becoming too opaque, ensuring they remain distinguishable. The RGB optical flow is processed by the TAE to produce a noised representation, $d_t$ (see Eq. 1).

Next, we modify the model to predict the joint distribution of appearance and motion. We achieve this by altering the architecture to a dual input-output format, where the model takes both a noised video, $x_t$, and a noised flow, $d_t$, and predicts both signals. This requires modifying two linear projection matrices, $\mathbf{W}_{in}$ and $\mathbf{W}_{out}$ (see Fig. 4(a)).

First, we extend the input projection $\mathbf{W}_{in}$ to take two inputs– the video and motion latents, $x_t, d_t$. This is done by adding $C_{\text{TAE}} \cdot p^2$ zero-rows to obtain a dual-projection matrix $\mathbf{W}_{in}^+ \in \mathbb{R}^{2 \cdot C_{\text{TAE}} \cdot p^2 \times C_{\text{DiT}}}$ such that at initialization, the network is equivalent to the pre-trained DiT, and ignores the added motion signal. Second, we extend $\mathbf{W}_{out}$ with an additional output matrix to obtain $\mathbf{W}_{out}^+ \in \mathbb{R}^{C_{\text{DiT}} \times 2 \cdot C_{\text{TAE}} \cdot p^2}$. The added layer extracts the motion prediction from the joint latent representation. Together, $\mathbf{W}_{in}^+$ and $\mathbf{W}_{out}^+$, alter the model to a dual input-output format that processes and predicts both appearance and motion.

As shown in Fig. 4(a), our modifications maintain the original latent dimensions of the DiT. Essentially, this requires the model to *learn a single unified latent representation*, from which both signals are predicted using a linear projection. Plugging the above into Eq. 4 we get,

$$\mathbf{u}^+([x_t, d_t], y, t; \theta') = \mathcal{M}([x_t, d_t] \cdot \mathbf{W}_{in}^+, y, t; \theta) \cdot \mathbf{W}_{out}^+,$$

where $[\bullet]$ denotes concatenation in the channel dimension, $\theta'$ denotes the extended model weights as specified above, and $\mathbf{u}^+ = [u^x, u^d]$ denotes the dual output, where the first channels represent the appearance (video) prediction, while the last ones represent the motion (optical flow) prediction.

Finally, we extend the training objective to include an explicit motion term, thus the objective from Eq. 3 becomes,

$$\mathcal{L} = \mathbb{E}_{[x_1, d_1], [x_0, d_0], y, t} \left[ ||\mathbf{u}^+([x_t, d_t], y, t; \theta') - \mathbf{v_t}^+||_2^2 \right],$$
(6)

where $\mathbf{v_t}^+ = [v_t^x, v_t^d]$ is calculated using Eq. 2. Note that while we only modify two linear layers, we jointly fine-tune all the weights in the network, to allow the model to learn the new target distribution.

At inference, the model generates both the video and its motion representation from noise. Note that we are mostly interested in the video prediction, whereas the motion prediction guides the model toward temporally plausible outputs.

### 4.3. Inner-Guidance

As previously observed (Ho & Salimans, 2022), conditioning a diffusion model on an auxiliary signal does not guarantee that the model will faithfully consider the condition.

Therefore, we propose to modify the diffusion score function to steer the prediction toward plausible motion.

In our setting, there are two conditioning signals: the prompt, $y$, and the *noisy intermediate motion prediction*, $d_t$. Notably, $d_t$ inherently depends on the prompt and model weights, as it is generated by the model itself. Consequently, existing approaches that assume independence between conditions and model weights (e.g., Brooks et al. (2023)), are not applicable in this setting (Sec. 2, App. A). To address this, we propose to *directly modify the sampling distribution*,

$$\tilde{p}_{\theta'}([x_t, d_t]|y) \propto$$
$$p_{\theta'}([x_t, d_t]|y)p_{\theta'}(y|[x_t, d_t])^{w_1}p_{\theta'}(d_t|x_t, y)^{w_2},$$
(7)

where $p_{\theta'}([x_t, d_t]|y)$ is the original sampling distribution, $p_{\theta'}(y|[x_t, d_t])$ estimates the likelihood of the prompt given the joint prediction, and $p_{\theta'}(d_t|x_t, y)$ estimates the likelihood of the noisy motion prediction. The latter is aimed at improving the model's motion coherence, as it maximizes the likelihood of the motion representation of the generated video. Using Bayes' Theorem, Eq. 7 is equivalent to,

$$p_{\theta'}([x_t, d_t]|y) \left( \frac{p_{\theta'}([x_t, d_t], y)}{p_{\theta'}([x_t, d_t])} \right)^{w_1} \left( \frac{p_{\theta'}([x_t, d_t], y)}{p_{\theta'}(x_t, y)} \right)^{w_2}$$
$$\propto p_{\theta'}([x_t, d_t]|y) \left( \frac{p_{\theta'}([x_t, d_t]|y)}{p_{\theta'}([x_t, d_t])} \right)^{w_1} \left( \frac{p_{\theta'}([x_t, d_t]|y)}{p_{\theta'}(x_t|y)} \right)^{w_2},$$

where we omit all occurrences of $p_{\theta'}(y)$ since $y$ is an external constant input. Next, we can translate this to the corresponding score function by taking the log derivative,

$$(1 + w_1 + w_2)\nabla_{\theta'} \log p_{\theta'}([x_t, d_t]|y)$$
$$-w_1\nabla_{\theta'} \log p_{\theta'}([x_t, d_t]) - w_2\nabla_{\theta'} \log p_{\theta'}(x_t|y).$$
(8)

Following Ho & Salimans (2022), we jointly train the model to be conditional and unconditional on both auxiliary signals, $y, d$ by randomly dropping out the text in 30% of the training steps, and the optical flow in 20% of the steps (setting $d = \mathbf{0}$). In particular, when the optical flow signal is dropped, the loss is derived with respect to the appearance only, resulting in the training scheme described in Sect. 4.1. Thus, the overall guidance formulation during inference becomes,

$$\tilde{\mathbf{u}}^+([x_t, d_t], y, t; \theta') = (1 + w_1 + w_2) \cdot \mathbf{u}^+([x_t, d_t]), y, t; \theta')$$
$$-w_1 \cdot \mathbf{u}^+([x_t, d_t], \emptyset, t; \theta') - w_2 \cdot \mathbf{u}^+([x_t, \emptyset], y, t; \theta').$$

Unless stated otherwise, all experiments use $w_1 = 5, w_2 = 3$, where $w = 5$ is the base model's text guidance scale.

## 5. Experiments

We conduct qualitative and quantitative experiments to demonstrate the effectiveness of VideoJAM. We benchmark our models against their base (pre-trained) versions, as well as leading proprietary and open-source video models, to highlight VideoJAM's enhanced motion coherence.

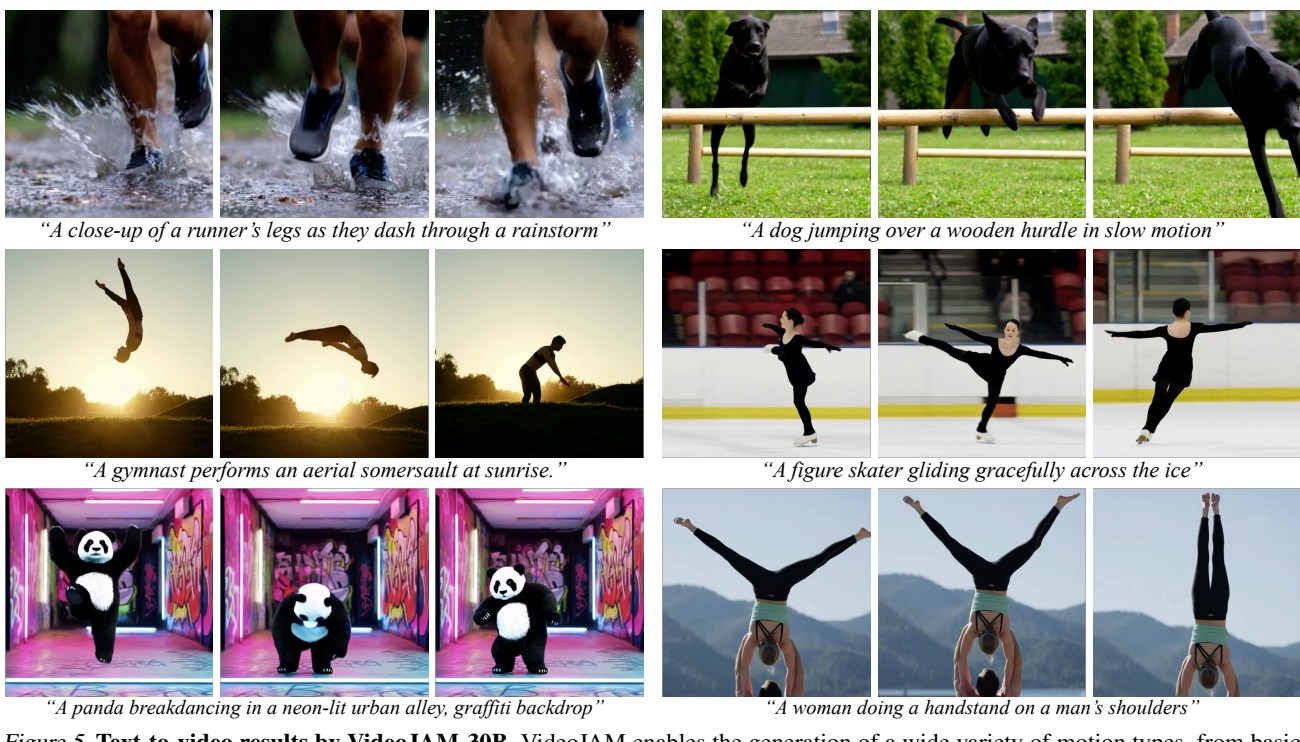

*Figure 5.* **Text-to-video results by VideoJAM-30B.** VideoJAM enables the generation of a wide variety of motion types, from basic motion (e.g., running) to complex motion (e.g., acrobatics), and improved physics (e.g., jumping over a hurdle).

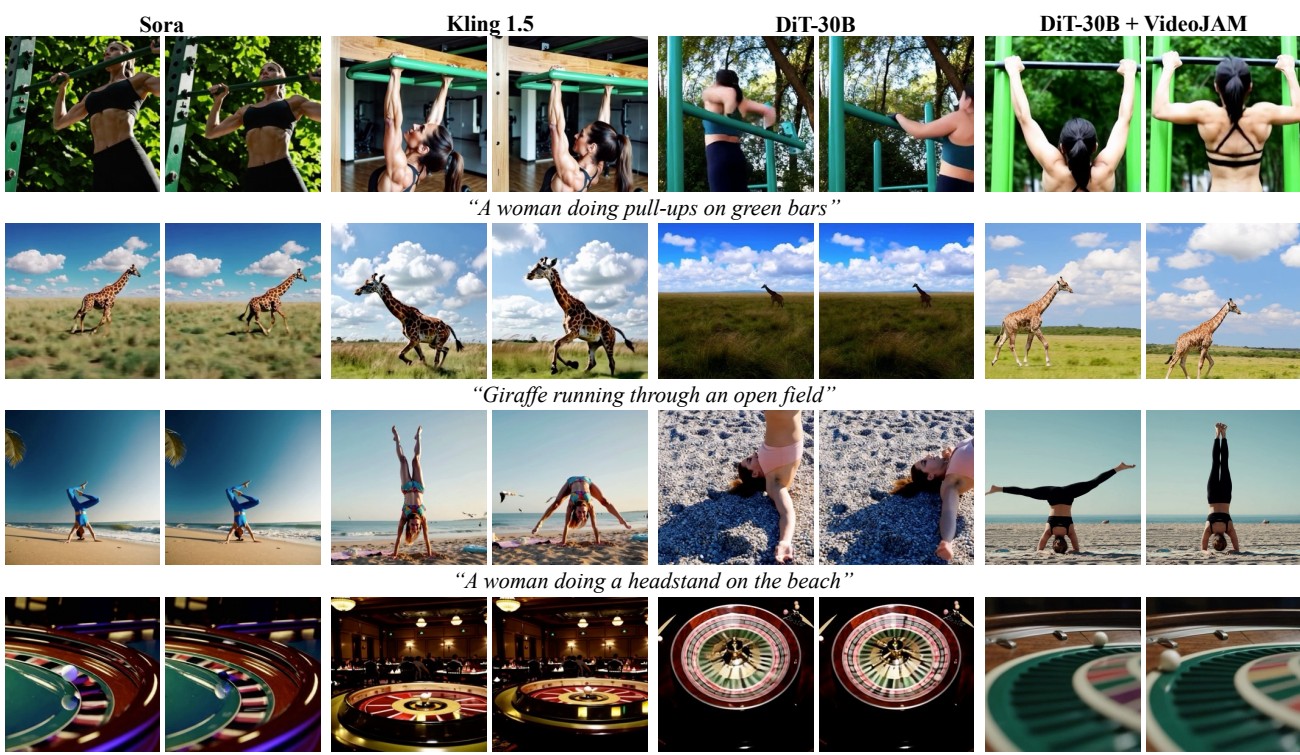

*Figure 6.* **Qualitative comparisons** between VideoJAM-30B and the leading baselines- Sora, Kling, and DiT-30B on representative prompts from VideoJAM-bench. The baselines struggle with basic motion, displaying "backward motion" (Sora, 2nd row) or unnatural motion (Kling, 2nd row). The generated content defies the basic laws of physics e.g., people passing through objects (DiT, 1st row), or objects that appear or evaporate (Sora, DiT, 4th row). For complex motion, the baselines display static motion or deformations (Sora, Kling, 1st, 3rd row). Conversely, in all cases, VideoJAM produces temporally coherent videos that better adhere to the laws of physics.

**Implementation Details** We consider two variants of the DiT text-to-video model, DiT-4B and DiT-30B, to demonstrate that motion coherence is a common issue for both small and large models. All of our models are trained with a spatial resolution of $256 \times 256$ for efficiency. The models are trained to generate 128 frame videos at 24 frames per second, resulting in 5-second video generations. Both DiT models were pre-trained using the framework in Sec. 4.1 on an internal dataset of $\mathcal{O}(100\,\text{M})$ videos. We then fine-tune the models with VideoJAM using 3 million random samples from the model's original training set, which constitute less than 3% of the training videos. This allows our fine-tuning phase to be light and efficient. During this fine-tuning, we employ RAFT (Teed & Deng, 2020) to obtain optical flow. For more implementation details, see App. C.

**Benchmarks** We use two benchmarks for evaluation. First, we introduce VideoJAM-bench, constructed specifically to test motion coherence. Second, we consider the Movie Gen (MGen) benchmark (Polyak et al., 2024) to show the robustness of our results.

VideoJAM-bench addresses limitations in existing benchmarks, including MGen, which do not fully evaluate real-world scenarios with challenging motion. For example, MGen's second-largest category, "unusual activity" (23.4% of MGen), contrasts with our objective of evaluating real-world ("usual") dynamics. The third largest category, "scenes" (19.9% of MGen), focuses on nearly static scenes in nature, thus inherently prioritizes appearance over meaningful motion. Even for categories that overlap with ours such as "animals", the representative example given by MGen is "a curious cat peering out from a cozy hiding spot".

To construct VideoJAM-bench, we consider prompts from four categories of natural motion that challenge video generators (see Fig. 2): basic motion, complex motion, rotational motion, and physics. We use a holdout set from our training data—on which no model was trained—and employ an LLM to select the top 128 prompts that best fit at least one of the four categories and describe a single, specific, and clear motion. To avoid biasing the evaluation toward a specific prompt style, we task the LLM with modifying the prompts to be of varying lengths and detail levels. A full list of our prompts can be found in App. D.

**Baselines** We consider a wide variety of state-of-the-art models, both proprietary and open-source. In the smaller category, we include CogVideo2B, CogVideo5B (Hong et al., 2022), PyramidFlow (Jin et al., 2024a), and the base model DiT-4B. In the larger category, we evaluate leading open-source models (Mochi (Genmo, 2024), CogVideo5B) and proprietary models with external APIs (Sora (Brooks et al., 2024), Kling 1.5 (KlingAI, 2024), RunWay Gen3 (RunwayML, 2024)), along with the base model DiT-30B. The leading baselines were selected using the video leaderboard.

*Table 1.* **Comparison of VideoJAM-4B with prior work on VideoJAM-bench.** Human evaluation shows *percentage of votes favoring VideoJAM*; automatic metrics use VBench.

| Method | Human Eval | | | Auto. Metrics | |
|---|---|---|---|---|---|
| | Text Faith. | Quality | **Motion** | Appearance | **Motion** |
| CogVideo2B | 84.3 | 94.5 | 96.1 | 68.3 | 90.0 |
| CogVideo5B | 62.5 | 74.7 | 68.8 | 71.9 | 90.1 |
| PyramidFlow | 76.6 | 83.6 | 82.8 | 73.1 | 89.6 |
| DiT-4B | 71.1 | 77.3 | 82.0 | **75.2** | 78.3 |
| **+VideoJAM** | - | - | - | 75.1 | **93.7** |

*Table 2.* **Comparison of VideoJAM-30B with prior work on VideoJAM-bench.** Human evaluation shows *percentage of votes favoring VideoJAM*; automatic metrics use VBench.

| Method | Human Eval | | | Auto. Metrics | |
|---|---|---|---|---|---|
| | Text Faith. | Quality | **Motion** | Appearance | **Motion** |
| CogVideo5B | 73.4 | 71.9 | 85.9 | 71.9 | 90.1 |
| RunWay Gen3 | 72.2 | 76.6 | 77.3 | 73.2 | 92.0 |
| Mochi | 56.1 | 65.6 | 74.2 | 69.9 | 89.7 |
| Sora | 56.3 | 51.7 | 68.5 | 75.4 | 91.7 |
| Kling 1.5 | 51.8 | 45.9 | 63.8 | **76.8** | 87.1 |
| DiT-30B | 71.9 | 74.2 | 72.7 | 72.4 | 88.1 |
| **+VideoJAM** | - | - | - | 73.4 | **92.4** |

### 5.1. Qualitative Experiments

Figures 1, 5, 9 present results obtained using VideoJAM-30B. The results demonstrate a wide variety of motion types that challenge existing models such as gymnastics (e.g., air splits), prompts that require physics understanding (e.g., fingers pressed into slime, basketball landing in a net), etc.

Figure 6 compares VideoJAM with the leading baselines, Sora and Kling, and the base model, DiT-30B, on prompts from VideoJAM-bench. The comparison highlights motion issues in state-of-the-art models. Even simple motions, such as a running giraffe (second row), show problems like "backward motion" (Sora) or unnatural movements (Kling, DiT-30B). Complex motions, like pull-ups or headstands, result in static videos (Sora, first and third rows; Kling, first row) or body deformations (Kling, third row). The baselines also exhibit physics violations, such as objects disappearing or appearing (Sora, DiT-30B, fourth row). In contrast, VideoJAM consistently produces coherent motion.

### 5.2. Quantitative Experiments

We evaluate appearance and motion quality, as well as prompt fidelity using both automatic metrics and human evaluations. In all our comparisons, each model runs *once* with the same random seed for all the benchmark prompts. For the automatic metrics, we use VBench (Huang et al., 2024), which assesses video generators across disentangled

*Table 3.* **Ablation study.** Ablations of the primary components of our framework on VideoJAM-4B using VideoJAM-bench. Human evaluation shows percentage of votes favoring VideoJAM.

| | Human Eval | | | Auto. Metrics | |
|---|---|---|---|---|---|
| Ablation type | Text Faith. | Quality | **Motion** | Appearance | **Motion** |
| w/o text guidance | 68.0 | 62.5 | 63.3 | 74.5 | 93.3 |
| w/o Inner-Guidance | 68.9 | 64.4 | 66.2 | **75.3** | 93.1 |
| w/o optical flow | 79.0 | 70.4 | 80.2 | 74.7 | 90.1 |
| IP2P guidance | 73.7 | 85.2 | 78.1 | 72.0 | 90.4 |
| **+VideoJAM-4B** | - | - | - | 75.1 | **93.7** |

axes. We aggregate the scores into two categories- appearance and motion, following the paper. The metrics evaluate the per-frame quality, aesthetics, subject consistency, the amount of generated motion, and motion coherence. More details on the metrics and their aggregation can be found in App. C.1.

For the human evaluations, we follow the Two-alternative Forced Choice (2AFC) protocol, similar to Rombach et al. (2022); Blattmann et al. (2023a), where raters compare two videos (one from VideoJAM, one from a baseline) and select the best one based on quality, motion, and text alignment. Each comparison is rated by 5 unique users, providing at least 640 responses per baseline for each benchmark.

The results of the comparison on VideoJAM-bench for the 4B, 30B models are presented in Tabs. 1, 2, respectively. Additionally, a full breakdown of the automatic metrics is presented in App. D. The results of the comparison on the Movie Gen benchmark are presented in App. E. In all cases, VideoJAM outperforms all baselines in all model sizes in terms of motion coherence, across both the automatic and human evaluations by a sizable margin (Tabs. 1, 2, 7).

Notably, VideoJAM-4B outperforms the CogVideo5B baseline, even though the latter is 25% larger. For the 30B variant, VideoJAM surpasses even proprietary state-of-the-art models such as Kling, Sora and Gen3 (63.8%, 68.5%, 77.3% preference in motion, respectively). These results are particularly impressive given that Video-JAM was trained at a significantly lower resolution (256) compared to the baselines (768 and higher) and fine-tuned on only 3 million samples. While this resolution disparity explains why proprietary models like Kling and Sora surpass ours in visual quality (Tab. 2), VideoJAM consistently demonstrates substantially better motion coherence.

Most critically, VideoJAM significantly improves motion coherence in its base models, DiT-4B and DiT-30B, in a direct apples-to-apples comparison. Human raters preferred VideoJAM's motion in 82.0% of cases for DiT-4B and 72.7% for DiT-30B. Raters also favored VideoJAM in quality (77.3%, 74.2% in 4B, 30B) and text faithfulness (71.1%, 71.9% in 4B, 30B), indicating that our approach

**(a) Motion observed in "zoom-out"**     **(b) Physics of object interactions**

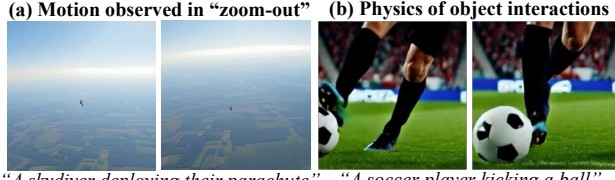

*"A skydiver deploying their parachute"*     *"A soccer player kicking a ball"*

*Figure 7.* **Limitations.** Our method is less effective for: (a) motion observed in "zoom-out" (the moving object covers a small part of the frame). (b) Complex physics of object interactions.

also enhances other aspects of the generation.

### 5.3. Ablation Study

We ablate the primary design choices of our framework. First, we ablate the use of text guidance and motion guidance in our inner guidance formulation (by setting $w_2 = 0$, $w_1 = 0$ in Eq. 8, respectively). Next, we ablate the use of motion prediction during inference altogether, by dropping the optical flow at each inference step ($d = \mathbf{0}$). Finally, we ablate our guidance formulation by replacing it with the InstructPix2Pix (IP2P) guidance (Brooks et al., 2023) (see Sec. 2, App. A). Note that the results of the DiT models in Tabs. 1, 2 also function as ablations, as they ablate the use of VideoJAM during training and inference.

The results are reported in Tab. 3. All ablations cause significant degradation in motion coherence, where the removal of motion guidance is more harmful than the removal of the text guidance, indicating that the motion guidance component indeed steers the model toward temporally coherent generations. Furthermore, dropping the optical flow prediction at inference is the most harmful, substantiating the benefits of the joint output structure to enforce plausible motion. The InstructPix2Pix guidance comparison is further indication that our Inner-Guidance formulation is most suited to our framework, as it gives the second lowest result in terms of motion.

Finally, note that human evaluators consistently prefer VideoJAM in terms of visual quality and text alignment over all the ablations, further establishing that VideoJAM benefits all aspects of video generation.

### 5.4. Limitations

While VideoJAM significantly improves temporal coherence, challenges remain (see Fig. 7). First, due to computational constraints, we rely on both limited training resolution and RGB motion representation, which hinder the model's ability to capture motion in "zoomed-out" scenarios where moving objects occupy a small portion of the frame. In these cases, the relative motion magnitude is reduced, making the representation less informative (Eq. 5). For example, in Fig. 7(a), no parachute is deployed, and the motion appears incoherent. Second, while motion and physics are

intertwined, leading to improved physics, our motion representation lacks explicit physics encoding. This limits the model's ability to handle complex physics of object interactions. For example, in Fig. 7(b), the player's foot does not touch the ball before it changes trajectory.

## 6. Conclusions

Video generation poses a unique challenge, requiring the modeling of both spatial interactions and temporal dynamics. Despite impressive advancements, video models continue to struggle with temporal coherence, even for basic motions well-represented in training datasets (Fig. 2). In this work, we identify the training objective as a key factor that prioritizes appearance fidelity over motion coherence.

To address this, we propose VideoJAM, a framework that equips video models with an explicit motion prior. The core idea is intuitive and natural: a single latent representation captures both appearance and motion jointly. Using only two additional linear layers and no additional training data, VideoJAM significantly improves motion coherence, achieving state-of-the-art results even against powerful proprietary models. Our approach is generic, offering numerous opportunities for future enhancement of video models with real-world priors such as complex physics, paving the way for holistic modeling of real-world interactions.

## Impact Statement

The primary goal of this work is to advance motion modeling in video generation, empowering models to understand and represent the world more faithfully. As with any technology in the content generation field, video generation carries the potential for misuse, a concern that is widely discussed within the research community. However, our work does not introduce any specific risks that were not already present in previous advancements. We strongly believe in the importance of developing and applying tools to detect biases and mitigate malicious use cases, ensuring the safe and fair use of generative tools, including ours.

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

# A. Compositional Guidance vs. Inner-Guidance

Liu et al. (2022) proposed *Composable Diffusion Models* where a diffusion model can be conditioned on several signals $c_1, \ldots, c_n$. The model's conditional sampling distribution is, therefore,

$$p_\theta(x|c_1, \ldots, c_n) = \frac{p_\theta(x, c_1, \ldots, c_n)}{p_\theta(c_1, \ldots, c_n)} \propto p_\theta(x, c_1, \ldots, c_n) \propto p_\theta(x) \prod_{i=1}^{n} p_\theta(c_i|x). \tag{9}$$

where $\theta$ represents the model weights, and $p$ is the sampling distribution. Importantly, this formulation assumes that $c_1, \ldots, c_n$ are *independent of each other and the weights of the model $\theta$*, allowing to drop the denominator $p_\theta(c_1, \ldots, c_n)$. Notice that this assumption does not hold in our setting, where the motion condition $d_t$ is noisy and strictly dependent on the neural network, as one of its outputs, as well as the text conditioning, as it serves as another input to the model.

Inspired by Liu et al. (2022), InstructPix2Pix (IP2P) (Brooks et al., 2023) used a similar compositional formulation to extend Classifier-Free Guidance (Ho & Salimans, 2022) to two conditioning signals. Formally, given two conditions $c_1, c_2$,

$$p_\theta(x|c_1, c_2) = \frac{p_\theta(x, c_1, c_2)}{p_\theta(c_1, c_2)} = \frac{p_\theta(c_1|c_2, x)p_\theta(c_2|x)p_\theta(x)}{p_\theta(c_1, c_2)}, \tag{10}$$

taking the log derivative this gives us,

$$\nabla \log p_\theta(x|c_1, c_2) = \nabla \log p_\theta(c_1|c_2, x) + \nabla \log p_\theta(c_2|x) + \nabla \log p_\theta(x) - \nabla \log p_\theta(c_1, c_2), \tag{11}$$

next, the IP2P formulation assumes (similar to Liu et al. (2022)) that we can omit the term $p_\theta(c_1, c_2)$ since it is independent of $\theta$, which is again incorrect in our case.

For completeness, our ablations in Sec. 5 compare our Inner-Guidance formulation with that of IP2P, and find that this theoretical gap causes significant degradation in the performance. The direct interpretation of Eq. 11 to VideoJAM employed in our experiments is as follows,

$$\tilde{\mathbf{u}}^+([x_t, d_t], y, t; \theta') = \mathbf{u}^+([x_t, \emptyset]), \emptyset, t; \theta')+$$
$$w_1 \cdot \left(\mathbf{u}^+([x_t, d_t], \emptyset, t; \theta') - \mathbf{u}^+([x_t, \emptyset]), \emptyset, t; \theta')\right)+$$
$$w_2 \cdot \left(\mathbf{u}^+([x_t, d_t], y, t; \theta') - \mathbf{u}^+([x_t, d_t], \emptyset, t; \theta')\right)$$

where the notations follow Sec. 4.3, and we employ the same guidance scales as we do for Inner-Guidance, i.e. $w_1 = 3, w_2 = 5$. Note that the notations for $w_1, w_2$ are reversed with respect to Eq. 8 since IP2P condition on the visual signal first and the textual signal second and order matters for IP2P, while our Inner-Guidance formulation is order invariant.

# B. Motivation Experiments

To exemplify that steps $t \leq 60$ of the generation are indeed meaningful to determine the motion, we conduct an SDEdit (Meng et al., 2022) experiment, in which we noise videos to different timesteps $(20, 60, 80)$, and continue the generation given the noised videos. In Fig. 8, we show a representative appearance frame and two motion frames for each video, using RAFT (Teed & Deng, 2020) to estimate optical flow. We observe that the coarse motion and structure of the generated videos are determined between steps 20 and 60, since the generation from step 20 changes the entire video while starting from step 60 maintains the coarse motion and structure of the input video, suggesting that they are already determined by the input noisy video. Note that the appearance may still change between steps 60 and 80 (right), whereas from step 80, both appearance and motion seem to be determined.

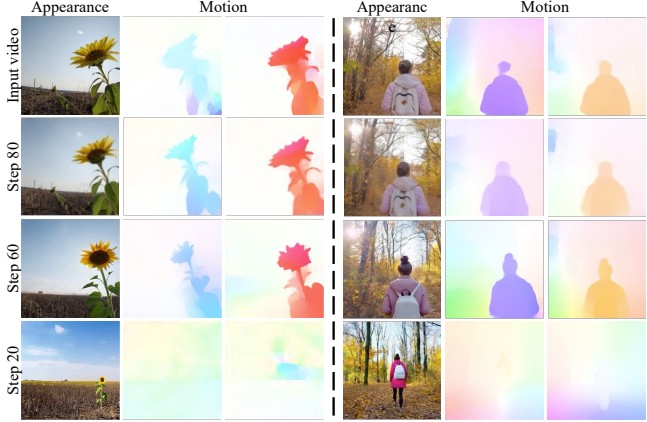

*Figure 8.* **Qualitative motivation.** We noise input videos to different timesteps $(20, 60, 80)$ and continue the generation. By step 60, the video's coarse motion and structure are mostly determined.

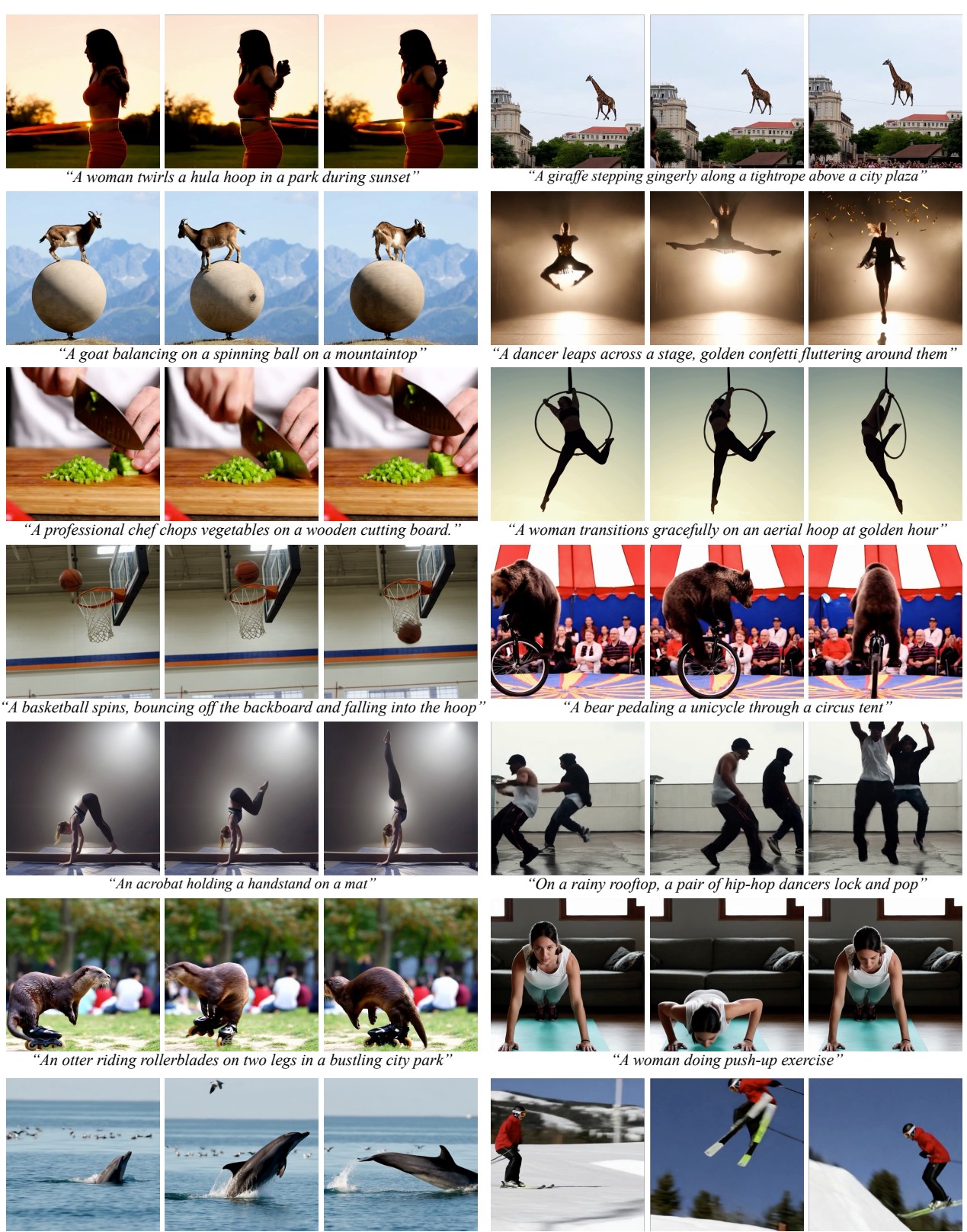

*"A woman twirls a hula hoop in a park during sunset"*

*"A giraffe stepping gingerly along a tightrope above a city plaza"*

*"A goat balancing on a spinning ball on a mountaintop"*

*"A dancer leaps across a stage, golden confetti fluttering around them"*

*"A professional chef chops vegetables on a wooden cutting board."*

*"A woman transitions gracefully on an aerial hoop at golden hour"*

*"A basketball spins, bouncing off the backboard and falling into the hoop"*

*"A bear pedaling a unicycle through a circus tent"*

*"An acrobat holding a handstand on a mat"*

*"On a rainy rooftop, a pair of hip-hop dancers lock and pop"*

*"An otter riding rollerblades on two legs in a bustling city park"*

*"A woman doing push-up exercise"*

*"A dolphin leaps out of the sea, a flock of birds flying in the background"*

*"A skier performing a spin midair"*

*Figure 9.* **Additional text-to-video results using VideoJAM-30B.**

# C. Implementation Details

VideoJAM-4B was fine-tuned using 32 A100 GPUs with a batch size of 32 for $50,000$ iterations on a spatial resolution of $256 \times 256$. It has a latent dimension of 3072 and 32 attention blocks (same as the base model). VideoJAM-30B was fine-tuned using 256 A100 GPUs with a batch size of 256 for $35,000$ iterations on a spatial resolution of $256 \times 256$. It has a latent dimension of 6144 and 48 attention blocks (same as the base model). Each attention block is constructed of a self-attention layer that performs spatiotemporal attention between all the video tokens, and a cross-attention layer that integrates the text. Both models were trained with a fixed learning rate of $5e-6$, using the Flow Matching paradigm (Lipman et al., 2023) (see Sec. 4.1).

During inference, we perform 100 denoising steps with a linear quadratic t-schedule using a text guidance scale of $w_1 = 5$ and a motion guidance scale of $w_2 = 3$ (see Eq. 8), other than the ablations that test these components. Additionally, we only employ the motion guidance for the first half of the generation steps (50 steps) following the conclusions from our motivational experiments (Sec. 3), as these are the steps that determine the coarse motion in the video, and display less sensitivity to temporal incoherence before applying VideoJAM. In practice, Inner-Guidance is performed similarly to Classifier-Free Guidance (Ho & Salimans, 2022), where all results are generated in a batch $\mathbf{u}^+([x_t, d_t]), y, t; \theta'), \mathbf{u}^+([x_t, d_t], \emptyset, t; \theta'), \mathbf{u}^+([x_t, \emptyset], y, t;)$ and the final prediction is calculated following Eq. 8. The models are trained to generate 128 frame videos at 24 frames per second, resulting in 5-second video generations.

The models operate in the latent space of a TAE, as specified in Sec. 4.1. The TAE structure follows that of Polyak et al. (2024), with a temporal compression rate of $\times 8$ and a spatial compression rate of $8 \times 8$. The Transformer patch size is $1 \times 2 \times 2$. The text prompt conditioning is processed by three different text encoders: UL2 (Tay et al., 2022), ByT5 (Xue et al., 2022), and MetaCLIP (Xu et al., 2023).

Both DiT models were pre-trained using the framework in Sec. 4.1 on a dataset of $\mathcal{O}(100\,\text{M})$ videos. We then fine-tune the models using VideoJAM on under 3 million random samples from the model's original training set, which constitute less than $3\%$ of the training videos. This allows our fine-tuning phase to be light and efficient. During this fine-tuning, we employ RAFT (Teed & Deng, 2020) to obtain optical flow per training video.

Since each of the baselines generates videos in different resolutions, we resize the baseline results to a 256 resolution to facilitate a fair and unbiased comparison. No cherry-picking is involved in the evaluation of any of the models, and the first result obtained by each model is taken. All baselines produce the same length of videos (5 seconds), therefore we only resize the videos spatially. For the qualitative results in the website, we train an additional super-resolution model to spatially upsample the $256 \times 256$ videos to $512 \times 512$ videos. The training regime follows that of VideoJAM-30B. Note that all our experiments (besides the visualizations on the website) are in the lower 256 resolution due to resource limitations.

## C.1. VBench Metrics

We employ all metrics supported by VBench on both VideoJAM-bench and the Movie Gen benchmark. Inspired by the protocol in the VBench paper, we split the metrics into a motion category and an appearance category. For the appearance category, we include the aesthetic quality and image quality metrics, which assess the per-frame quality of the generated videos, as well as subject consistency and background consistency, which assess the model's ability to maintain a consistent appearance. For motion comprehension, we include the motion smoothness score, which aims to assess the realism of the motion, and the dynamic degree score which estimates the amount of motion in the generated videos. In other words, the motion score measures the model's ability to generate meaningful motion (i.e., non-static videos) that is also coherent and plausible.

All scores are normalized and a weighted score is calculated according to the weights suggested in the VBench paper. The full results of all VBench metrics for each benchmark are reported in App. D, E.

# D. VideoJAM-bench: Automatic Metrics Breakdown and Prompts

In the following, we provide a breakdown of the automatic metrics calculated on our motion benchmark using VBench (Huang et al., 2024) for the 4B model (Tab. 4) and the 30B model (Tab. 5). As mentioned in App. C.1, the motion metrics measure the amount of motion in the video and the coherence of the motion. In the smaller model category, CogVideo2B scores the highest dynamic degree and the lowest motion smoothness. This indicates that while there is abundant motion in the generated videos, it is incoherent. The DiT-4B base model obtains the best smoothness score, and

the worst dynamic degree, indicating that it produces videos with very subtle movements. As can be observed, VideoJAM strikes the best balance, where plenty of motion is generated while maintaining strong coherence.

For the larger DiT-30B model, we observe, again, that there is a trade-off between the dynamic degree and the motion smoothness, where CogVideo5B produces the most motion, yet it is incoherent. Among the competitive proprietary baselines, notice that Runway Gen 3 obtains a very high dynamic degree, yet it has the lowest motion smoothness among all the proprietary baselines (Runway Gen 3, Sora, Kling 1.5). In Fig. 5, we show comparisons to Sora and Kling since these are the most competitive with VideoJAM according to the human evaluation, which is generally considered to be a more reliable evaluation form (BarTal et al., 2024; Polyak et al., 2024; Wang et al., 2024). However, in the website, we include a comparison to Runway Gen 3 in addition to Sora and Kling for completeness. Furthermore, Kling shows the best motion smoothness, with the lowest dynamic degree. Observe that VideoJAM, again, strikes the best balance between motion coherence and the amount of generated motion. Additionally, it outperforms the base model (DiT-30B) across all motion metrics, and nearly all appearance metrics, indicating that our method improves all aspects of the generation.

For completeness, Tab. 6 presents the VBench metrics not supported for external benchmarks, calculated on the VBench prompts. The reported results demonstrate that VideoJAM improves almost all aspects of generation, indicating once again that our framework does not sacrifice video quality for improved temporal coherence.

A full list of the prompts considered in our motion benchmark is provided in App. F.

*Table 4.* **Breakdown of the automatic metrics** from VBench comparing our 4B model and previous work on VideoJAM-bench. Our method strikes the best balance between the dynamic degree (higher implies more motion) and the motion smoothness (higher implies smooth motion).

| Method | Appearance Metrics | | | | Motion Metrics | |
| --- | --- | --- | --- | --- | --- | --- |
| | Aesthetic Quality | Image Quality | Subject Consistency | Background Consistency | **Motion Smoothness** | **Dynamic Degree** |
| CogVideo2B | 46.9 | 48.9 | 87.8 | 93.9 | 97.1 | **88.6** |
| CogVideo5B | 51.1 | 52.9 | 91.3 | 95.3 | 97.3 | 87.5 |
| DiT-4B | **51.8** | **61.4** | 93.0 | **96.7** | **99.3** | 38.3 |
| **+VideoJAM-4B** | 51.6 | 61.1 | **93.5** | **96.7** | 98.8 | 87.5 |

*Table 5.* **Breakdown of the automatic metrics** from VBench comparing our 30B model and previous work on VideoJAM-bench. Our method strikes the best balance between the dynamic degree (higher implies more motion) and the motion smoothness (higher implies smooth motion).

| Method | Appearance Metrics | | | | Motion Metrics | |
| --- | --- | --- | --- | --- | --- | --- |
| | Aesthetic Quality | Image Quality | Subject Consistency | Background Consistency | **Motion Smoothness** | **Dynamic Degree** |
| CogVideo5B | 51.1 | 52.9 | 91.3 | 95.3 | 97.3 | **87.5** |
| RunWay Gen3 | 55.1 | 55.1 | 90.7 | 95.2 | 98.4 | 84.4 |
| Mochi | 49.5 | 48.8 | 89.7 | 95.2 | 98.4 | 78.1 |
| Sora | 56.8 | 57.7 | 93.0 | 96.4 | 98.7 | 82.0 |
| Kling 1.5 | **58.5** | **60.4** | **93.9** | **96.5** | **99.2** | 64.8 |
| DiT-30B | 49.2 | 56.8 | 91.3 | 95.5 | 98.8 | 71.1 |
| **+VideoJAM-30B** | 51.2 | 55.9 | 93.0 | 96.1 | 99.0 | 82.3 |

# E. Movie Gen Benchmark

We employ the prompts from the official benchmark labeled as containing "high" motion since our primary objective is to estimate motion coherence. Additionally, since the Movie Gen benchmark is significantly larger than VideoJAM-bench, and mostly contains less relevant prompts (Sec. 5), we consider the baselines that provide open-source code and can run automatically. Importantly, note that the apples-to-apples comparison to the pre-trained model, DiT-30B is presented for this benchmark as well, allowing us to assess the direct impact of VideoJAM on a large video generation model.

*Table 6.* **Breakdown of the remaining automatic metrics** from VBench comparing our 30B model with its base model, DiT-30B, on VBench prompts. Our method improves almost all metrics, indicating that VideoJAM does not compromise appearance for motion enhancement.

| Method | Temporal Flickering | Object Class | Multiple Objects | Human Motion | Color | Spatial Relationship | Scene | Appearance Style | Temporal Style | Overall Consistency |
|---|---|---|---|---|---|---|---|---|---|---|
| DiT-30B | **99.7** | 89.1 | 63.0 | 97.0 | 81.2 | 72.0 | **52.1** | 22.1 | 24.2 | 27.5 |
| +VideoJAM-30B | **99.7** | **90.7** | **70.0** | **99.0** | **91.1** | **73.3** | 49.7 | **23.3** | **24.4** | **27.6** |

The results are reported in Tab. 7, with a breakdown of the automatic metrics in Tab. 8. Similarly to the results on our motion benchmark, VideoJAM strikes the best balance between the amount of motion and the coherence of the generated motion. While CogVideo5B consistently produces the most motion, it is also consistently the least coherent baseline. Mochi, on the other hand, suffers from the complementary problem where less motion is generated. Notably, VideoJAM outperforms all baselines, by a significant margin across all metrics, both human-based and automatic (other than the dynamic degree, where CogVideo5B scores the highest, as mentioned). Importantly, we observe a consistent improvement over the base model used by VideoJAM, DiT-30B in both the appearance and motion metrics across all evaluations, which further substantiates our method's ability to improve all aspects of video generation.

*Table 7.* **Comparison of VideoJAM-30B with prior work on the Movie Gen benchmark.** Human evaluation shows *percentage of votes favoring VideoJAM*; automatic metrics use VBench.

| Method | Human Eval | | | Auto. Metrics | |
|---|---|---|---|---|---|
| | Text Faith. | Quality | **Motion** | Appearance | **Motion** |
| CogVideo5B | 61.4 | 77.0 | 78.7 | 70.8 | 88.8 |
| Mochi | 53.5 | 59.4 | 69.1 | 70.4 | 85.1 |
| DiT-30B | 60.3 | 64.6 | 66.1 | 70.5 | 87.3 |
| **+VideoJAM-30B** | - | - | - | **73.7** | **90.8** |

*Table 8.* **Breakdown of the automatic metrics** from VBench comparing our 30B model and previous work on the Movie Gen benchmark. Our method strikes the best balance between the dynamic degree (higher implies more motion) and the motion smoothness (higher implies smooth motion).

| Method | Appearance Metrics | | | | Motion Metrics | |
|---|---|---|---|---|---|---|
| | Aesthetic Quality | Image Quality | Subject Consistency | Background Consistency | **Motion Smoothness** | **Dynamic Degree** |
| CogVideo5B | 50.9 | 51.9 | 89.5 | 94.7 | 97.5 | **81.6** |
| Mochi | 50.4 | 50.1 | 89.0 | 95.4 | 98.9 | 60.7 |
| DiT-30B | 48.7 | 50.6 | 90.8 | 95.3 | 98.9 | 67.8 |
| **+VideoJAM-30B** | **51.5** | **56.4** | **93.3** | **96.2** | **99.1** | 76.9 |

## F. VideoJAM-bench Prompts

Below, we present the full set of 128 prompts used in our motion benchmark, VideoJAM-bench. The benchmark is designed to be diverse, encompassing simple motions (e.g., walking), complex human movements (e.g., gymnastics), rotational motions (e.g., spinning balls), and physics-based actions (e.g., a woman hula hooping). To ensure clarity, the prompts were refined using an LLM to focus on specific motion types, enabling a precise evaluation of the model's ability to generate coherent movement. Additionally, the prompts vary in detail and include camera instructions to test the model's performance across a wide range of scenarios.

1. *"A woman performing an intricate dance on stage, illuminated by a single spotlight in the first frame. She is dressed in a long black dress and a wide-brimmed hat, with her arms raised above her head. The woman dance Argentine*

*flamenco dance.”*

2. *“A woman doing a headstand on a beach.”*

3. *“A woman engaging in a challenging workout routine, performing pull-ups on green bars.”*

4. *“Two ibexes navigating a rocky hillside. They are walking down a steep slope covered in small rocks and dirt. In the background, there are more rocks and some greenery visible through an opening in the rocks.”*

5. *“A close-up of a runner’s legs as they sprint through a crowded city street, dodging pedestrians and street vendors, with the sounds of the city all around.”*

6. *“Athletic man doing gymnastics elements on horizontal bar in city park. Male sportsmen perform strength exercises outdoors.”*

7. *“A small dog playing with a red ball on a hardwood floor.”*

8. *“A woman engaging in a lively trampoline workout. The woman jumps and exercises on the trampoline. The background is a room with white walls and a white ceiling, and there are two large windows on the left side of the wall, and a mirror on the right side reflecting the womans image.”*

9. *“A man performing a handstand on a wooden deck overlooking a green lake surrounded by trees.”*

10. *“Young adult female performs an air gymnastic show on circus arena, holding ring in hand, making twine exercise, spin around”*

11. *“A woman enjoying the fun of hula hooping.”*

12. *“A man juggling with three red balls in a city street.”*

13. *“A white kitten playing with a ball.”*

14. *“A slow-motion shot captures a runner’s legs as they dash through a busy intersection, dodging cars and pedestrians, the city life bustling around them.”*

15. *“A young girl playing basketball in a red brick wall background. The girl, with fair skin and long blonde hair, is wearing a green jacket and has her left arm up to throw the ball. In the mid-frame, the girl is still playing basketball, with her right hand holding the ball in front of her face. The ground is dark gray cement with some patches of grass growing through it. As the video progresses, the girl is seen playing near some grassy areas on the ground.”*

16. *“A basketball game in progress, with two players reaching up to grab the ball as it spills out of the net. The player on the left has his hand outstretched, while the player on the right has both hands raised high. The ball is just above their fingertips, indicating that they are both trying to grab it simultaneously. The background of the image is blurred, but it appears to be a gymnasium or sports arena, with fluorescent lights illuminating the scene. As the video progresses, the players continue to jump and stretch to gain possession of the ball, their movements becoming more urgent and intense. The ball flies back and forth between them, with neither player able to secure it. In the final frame, the ball is still in mid-air, the players hands reaching up to grab it as the video ends.”*

17. *“A group of basketballs floating in mid-air in slow motion, with a larger ball on the left and two smaller balls on either side in the initial frame. Overall, the video captures the dynamic and energetic movement of basketballs as they float and bounce through space.”*

18. *“A dog playing with an orange ball with blue stripes. The dog picks up the ball and holds it in its mouth, conveying a sense of playfulness and energy. Throughout the video, the dog is seen playing with the ball, capturing the joy and excitement of the moment.”*

19. *“A woman doing acrobatic exercises on a pole in the gym.”*

20. *"A young man performing a cartwheel on a gray surface. He is dressed in orange pants, a black t-shirt, and white sneakers. As he executes the cartwheel, his right arm is extended upward, and his left arm is bent at the elbow, reaching down to the ground. His right leg is extended behind him, while his left leg is bent at the knee, pointing towards the camera. The background is a featureless gray wall. The mans energy and focus are evident as he completes the cartwheel, showcasing his athleticism and coordination."*

21. *"A golden retriever playing fetch on a grassy field. The dog is running with a frisbee in its mouth, its fur waving in the wind."*

22. *"A brightly colored ball spins rapidly on a flat surface, its patterns blurring as it twirls in place."*

23. *"A basketball spins on a player's fingertip, maintaining balance while gradually slowing down."*

24. *"A person jogs along a forest trail at dawn, their feet kicking up dirt with every stride, the sunlight filtering through the trees casting long shadows on the path."*

25. *"A child jumps up and down in place, their feet leaving the ground briefly before landing again."*

26. *"A person lifts one knee high in a marching motion, then places their foot back down and repeats with the other leg."*

27. *"Professional cyclist training indoors on a stationary bike trainer."*

28. *"Young Adult Male Doing Handstand on the beach."*

29. *"A young woman practicing boxing in a gym."*

30. *"A man jumping in a pool."*

31. *"A man doing push-ups on a ledge overlooking a body of water. The man appears to be doing a push-up, with his head down."*

32. *"A man enjoying a leisurely bike ride along a road next to a body of water during a sunset. As he pedals, he looks down at his front wheel, seemingly focused on his ride. The background features a large body of water, with a gray wall along the left side of the road in the mid-frame caption."*

33. *"close up shot of the feet of a woman exercising on a cardio fitness machine in a fitness club. As the video progresses, the legs continue to pedal the bike in a smooth, consistent motion."*

34. *"A woman engaging in an intense workout on a stationary bike while monitoring her progress on a screen."*

35. *"A woman running along a river with a city skyline in the background."*

36. *"A skier walking up a snowy hill with their skis on their back and ski poles in hand."*

37. *"A woman running through a grassy area, wearing a black tank top, gray and white leggings, and white sneakers. She is initially running on a dirt path, surrounded by trees with green leaves. As she continues to run, the scenery changes to a park, and her leggings change to a blue and white pattern. She is still running on a dirt path, surrounded by trees and green grass. The video captures her journey as she runs through the grassy area, enjoying the outdoors and the beauty of nature."*

38. *"A young girl coloring at her desk."*

39. *"A close-up of a runner's legs as they dash through a rainstorm, their shoes splashing through puddles as they push forward with determination."*

40. *"Tracking camera shot. A kangaroo hops swiftly across an open grassy plain."*

41. *"A close-up view of a spiral object with a glowing center. The object appears to be made of metal and has a shiny, reflective surface. . This light creates a series of concentric circles around the objects circumference, which are visible due to the reflection of the light off the metal surface."*

42. *"A roulette wheel in a dimly lit room or casino floor. In the center of the wheel, there's a small white ball that appears to be spinning rapidly as it moves around the track. The ball spins around the wheel, and the wheel rotates counterclockwise."*

43. *"A close-up of a jogger's feet as they run along a rocky coastal path, their shoes gripping the uneven surface, with the ocean waves crashing below."*

44. *"A person's hands as they shape and mold clay on a pottery wheel. The hands are covered in brown clay and are visible from the elbows down, with the forearms resting on top of a large yellow pottery wheel."*

45. *"A conveyor belt pouring out a large amount of small, brown objects into a pile on the ground. The objects being poured are falling from the conveyor belt in a steady stream, forming a large pile on the ground below. In the background, the sky is bright blue and cloudless, providing a stark contrast to the darker colors of the conveyor belt and the pile of objects."*

46. *"A 3d rendering of coins and small objects floating against a black background. The coins are gold, silver, bronze, and copper, with various denominations and sizes. Some have a shiny finish, while others are matte or tarnished. The scene is chaotic and dynamic, with the objects seemingly flying around in all directions. As the video progresses, the coins and objects tumble and spin, creating a sense of movement and energy. By the end, the screen is filled with white objects of various shapes and sizes, suggesting that something exciting is happening."*

47. *"A puppy runs through a grassy field."*

48. *"A cinematic shot of a person walking along a quiet country road, their feet crunching on the gravel with every step, fields of wheat swaying in the breeze on either side."*

49. *"A washing machine undergoing a full cycle. It begins with a top-down view of the machine filled with water and white soap suds, with two black rubber seals on either side of the stainless steel drum. The video progresses to show the drum spinning, with the suds becoming more agitated and the seals moving along with the drums motion."*

50. *"Sweet Cherries on Stems Colliding and Splashing Water Droplets"*

51. *"A series of colorful balloons floating in mid-air, creating a festive and celebratory atmosphere."*

52. *"A cinematic shot of a person jogging along a riverside path, their feet rhythmically tapping against the ground, the river flowing gently beside them."*

53. *"A green helicopter taking off from an airport runway."*

54. *"A hand holding a yellow fidget spinner. The hand is fair-skinned and holds the bright yellow fidget spinner with silver bearings. The background is blurred and appears to be trees against a blue sky. The video captures the subtle movements of the hand as it spins the fidget spinner, creating a soothing and mesmerizing visual effect. As the video progresses, the hand continues to hold the fidget spinner, showcasing its smooth and satisfying motion. The background remains blurred, adding a sense of tranquility to the scene. Overall, the video is a calming and enjoyable display of the simple pleasure of fidget spinning."*

55. *"A windmill spinning in a green field."*

56. *"A bicycle wheel spins forward, moving in a circular motion while keeping balance."*

57. *"A waterwheel turns as water flows over it, the paddles rotating consistently."*

58. *"A close-up of a person's legs as they walk through a sun-dappled forest, the light playing off their shoes as they navigate the uneven terrain."*

59. *"A man riding a mountain bike on a dirt trail."*

60. *"A child's toy top spins on a smooth surface, rotating without stopping."*

61. *"A basketball spins on a player's fingertip, showcasing balance and skill."*

62. *"A jellyfish swimming in shallow water. The jellyfish has a translucent body with a distinctive pattern of white circles and lines. It appears to be swimming just below the surface of the water, which is dark and murky due to the presence of algae or other aquatic plants."*

63. *"A cinematic shot of a person walking along a cobblestone street in a historic town, their feet making a rhythmic tap on the stones as they move."*

64. *"A group of horses grazing in a grassy field behind a black wooden fence"*

65. *"A fish swims forward in a steady line, its tail swaying side to side as it propels itself."*

66. *"A penguin waddles in a straight line, shifting from one foot to the other."*

67. *"A man is jumping rope on the sandy beach, with waves crashing in the background."*

68. *"A man enjoying water skiing on a brown river with a green shore and lily pads in the background. Water sprays up from underneath him as he skis across the surface of the lake."*

69. *"A man is swimming in a clear blue pool, enjoying the cool water and the freedom of movement in the pool. As he continues to swim, he glides gracefully through the water, his arms and legs moving in a smooth and coordinated rhythm."*

70. *"A kid running in the mountains of Campo Imperatore, Italy, at the sunset. He is wearing a red polo shirt, blue jeans, and brown shoes. As he runs, he passes by some white rocks on the ground."*

71. *"A woman doing push-up exercise on a beach at sunset."*

72. *"A woman is shown running through a field, with tall grass and wildflowers all around her. She is a fair-skinned woman with long, red hair, wearing a black t-shirt and leggings, and listening to music on her phone. In the background, there are trees and more fields of greenery."*

73. *"A man exercising with battle ropes at a gym."*

74. *"A person engaging in a boxing workout at a gym."*

75. *"A dark gray horse running in an enclosed corral. It is running towards the camera."*

76. *"A close-up of a runner's legs as they dash up a flight of stairs in a city park, their feet hitting each step with precision and power."*

77. *"A man is swimming in the ocean. In the background, the sky is hazy and overexposed, with the sun shining brightly above the horizon. As the video progresses, the man continues to swim, his arms moving rhythmically through the water."*

78. *"A herd of white cows walking down a dirt path. The cows are all facing forward and walking towards the right side of the image. The background is blurry but appears to be a field or pasture."*

79. *"A person jogs along a trail in a dense forest, their legs pumping as they navigate the roots and rocks that dot the path."*

80. *"A young woman dances in the night bustle against the backdrop of a glowing fanfare."*

81. *"A man is walking down the street while pushing a trash can. The man, wearing a red t-shirt, blue jeans, and brown sandals, pushes the black trash can on wheels."*

82. *"A man enjoying a mountain biking adventure through a forest. He is seen riding a black and white mountain bike down a dirt path, with his back to the camera."*

83. *"Women's legs walk into the sea with waves."*

84. *"A young man walking on a treadmill. He is wearing a white tank top and red shorts, and has his hands on the sides of the machine as he runs."*

85. *"Closeup of feet of a professional soccer player training with ball on stadium field with artificial turf."*

86. *"A helicopter flying over a forest. The helicopter is black and has two large rotor blades on top. It is flying low to the ground, with its nose pointing slightly upwards."*

87. *"A close-up of a person's feet as they walk through a field of wildflowers, their shoes brushing against the blooms with each step."*

88. *"A man is playing basketball, dribbling the ball and making shots."*

89. *"A giraffe running through an open field. The background is a bright blue sky with fluffy white clouds."*

90. *"A person jogs along a city waterfront, their legs moving steadily as the sun sets, casting a warm glow over the water and the buildings behind them."*

91. *"A woman is doing push-ups on a mat in the studio."*

92. *"Two dancers perform on a stage. The man stands behind the woman with his left arm is lifted over his head and the other is stretched to the right. The woman lets go of the man's right hand, swinging her leg to the left and performing a pirouette. She spins four times and ends up facing the man."*

93. *"A woman drinks from a water bottle in a forest. The woman has fair skin and brown hair. She is wearing a black jacket and black and white gloves."*

94. *"Tracking camera shot. A polar bear walks across a snowy landscape. It looks curiously around as it plods through the snow. The background is a snowy landscape with footprints visible in the snow. Sunlight shines from overhead and casts the bear's shadow on the snow."*

95. *"A cinematic shot of a person walking through a desert at midday, their legs moving slowly but steadily across the sand dunes, with heat waves distorting the distant horizon."*

96. *"A man jumping rope on a dark stage. His movements are fluid and energetic. Two spotlights shine down from above him."*

97. *"A woman twirls a hula hoop around her waist in a park during sunset. The woman, with medium-length curly black hair and a yellow tank top, stands on a grassy field surrounded by trees. As the hoop revolves around her waist, she shifts her hips rhythmically to keep it moving. The golden sunlight casts a long shadow behind her."*

98. *"A man exercises on a leg press machine at a gym."*

99. *"A young woman enjoys a cup of coffee on a balcony."*

100. *"A man energetically bangs on a drum kit. He holds drumsticks in both hands and bashes on the drum kit with the drumsticks."*

101. *"A woman performs high knees on a beach."*

102. *"Aerial tracking camera shot. A white semi-truck drives on a highway."*

103. *"A woman is holding a clear wine glass partly filled with a burgundy-colored wine. Facing forward, the woman smiles, she raises the glass with her left hand and takes a small sip."*

104. *"A man works on a piece of wood in a workroom. He holds a shiny silver chisel with a wooden handle in his right hand."*

105. *"Sliced green apples are tossed in a brown liquid. The apples are cut into thick slices and have shiny green skins with some light-colored speckling. They begin to rotate clockwise, flying out in every direction as the light amber liquid splashes and swirls behind them."*

106. *"A baboon eats a mango."*

107. *"A young woman vapes in the living room. The woman exhales the thick, billowing smoke."*

108. *"A woman performing an aerial hoop trick. The woman hangs from a black aerial hoop attached to the ceiling by a rope. In the initial frame, she has her legs wrapped around the hoop and her arms extended outward, holding onto the hoop with both hands. Her body is twisted, looking up towards the ceiling, with her shadow cast on the white wall behind her. As the video progresses, she continues to hang from the hoop, her body twisted in various positions, her arms and legs wrapped around the hoop as she performs the aerial trick. The background remains the same, with shadows from the aerial hoop and the woman's body on the white wall."*

109. *"Modern urban street ballet dancer performing acrobatics and jumps."*

110. *"A woman doing a pirouette in an empty dance studio."*

111. *"A woman dancing hip hop, street dancing in the studio. Slow motion."*

112. *"A brunette woman doing some acrobatic elements on aerial hoop outdoors."*

113. *"A woman, with long brown hair and wearing a black top and gray bottoms, climbs on a pole with her right leg wrapped around it and her left arm extended upward. The background is a white wall with a mirror reflecting the woman's images."*

114. *"A man performing a backflip. Slow motion."*

115. *"A woman dancing in a gym. The woman is spinning around repeatedly."*

116. *"A group of duck are walking in a row, one after the other. The background is a Japanese temple."*

117. *"Arc camera shot. A young woman doing stretches on a beach."*

118. *"A woman walking through a field of beautiful sunflowers. She spins counterclockwise and laughs. A field of shoulder-length sunflowers grow in the background, with trees on the horizon stretching up towards a cloudy sky."*

119. *"Arc camera shot. A man playing the guitar."*

120. *"A boy blowing out candles on a birthday cake."*

121. *"A cheetah running in the Savannah."*

122. *"Tracking shot. A golden retriever runs through a grassy park. The dog's ears flop up and down with each bounding step, and its tongue hangs out to one side. A frisbee flies into view from the left, and the dog leaps into the air to catch it. A group of people in the background claps and cheers."*

123. *"A young girl skips down a quiet suburban street lined with trees. She has light brown skin and long, wavy black hair tied back with a red ribbon. The girl wears a white t-shirt, a denim skirt, and bright yellow sneakers. Her arms swing loosely as she skips"*

124. *"A woman doing sit-ups at a gym."*

125. *"A child riding his bicycle on a dirt path. The background is a dirt path lined with trees on either side."*

126. *"A runner moves at full speed along a suburban sidewalk. The background is rows of houses and trees passing by in a blur."*

127. *"A young woman engaging in a boxing workout. She is wearing red boxing gloves and a white t-shirt, and has long blonde hair. In the first frame, she is standing in front of a black punching bag, with her right arm extended and her left arm bent, ready to punch the bag. She appears focused and determined. In the second frame, she has moved to the left of the bag and is looking towards the right side of the image. She continues to punch the bag with her right arm extended and her left arm bent. In the final frame, she is still standing to the left of the bag and is looking towards the right side of the image. She is still wearing her red boxing gloves and white t-shirt, and her long blonde hair is visible. The background of a blue wall with a window on the left and a doorway on the right, as well as two black objects hanging from the ceiling. Throughout the video, the woman is intensely focused on her workout, punching the bag with precision and skill."*

128. *"A brown bear walks in a grassy field."*

