# OpenReview forum: "VideoJAM: Joint Appearance-Motion Representations for Enhanced Motion Generation in Video Models"
_ICML.cc/2025/Conference — ICML 2025 oral_

### Official Review · Reviewer_VpZA · 2025-03-13

**Overall Recommendation:** 4

**Summary:**

The paper proposes a joint appearance-motion learning framework for video generation. The authors are motivated by the key observation that the common pixel-based training objective is invariant to temporal perturbations. Therefore, they propose to equip the model with an explicit motion learning objective via a joint representation learning framework. In addition, the authors propose an inner-guidance mechanism that steers the prediction towards more coherent motion. The proposed method is extensively evaluated on VideoJAM-Bench and shows advantageous performance especially in motion coherence and physics following.

## update after rebuttal
The authors have addressed all of my concerns with extensive experimental results. Therefore, I am keeping my score of Accept.

**Claims And Evidence:**

The paper's claims are well supported by extensive empirical evidence. In particular, the experiment in Section 3 is insightful and effectively justifies the motivation of the paper.

**Essential References Not Discussed:**

Several related works on using motion representation for video generation are missing [Tulyakov'18, Shen'24, Jin'24].

---

[1] Tulyakov, et al. MoCoGAN: Decomposing Motion and Content for Video Generation. CVPR 2018.

[2] Shen, et al. Decouple Content and Motion for Conditional Image-to-Video Generation. AAAI 2024.

[3] Jin, et al. Video-LaVIT: Unified Video-Language Pre-training with Decoupled Visual-Motional Tokenization. ICML 2024.

**Experimental Designs Or Analyses:**

The experimental designs are quite comprehensive, but could be improved by considering scalability:

* **W2**: The most counterintuitive result is that VideoJAM-30B is inferior to VideoJAM-4B in terms of quality in human evaluation and all automatic metrics, according to Tables 1 and 2. This casts doubt on the scalability of the proposed method, since a well-known bitter lesson is that the introduction of human prior knowledge can reduce scalability. It would be better for the authors to explain the performance drop and provide additional results to prove scalability.

**Methods And Evaluation Criteria:**

The proposed method is suitable for video generation because of its ability to enhance motion coherence. The evaluation criteria are extensive, including human evaluation and automatic metrics on VideoJAM-Bench and Movie Gen benchmark, but there is room for improvement:

* **W1**: It would be better to evaluate on the original VBench dataset, as this would allow comparison with more open-source baselines on more general prompts.

**Other Comments Or Suggestions:**

Given the improved physics generation capability, it would be interesting to quantitatively evaluate it on recent physics-focused benchmarks such as PhyGenBench [Meng'24], Physics-IQ Benchmark [Motamed'25], WorldModelBench [Li'25].

---

[1] Meng, et al. Towards World Simulator: Crafting Physical Commonsense-Based Benchmark for Video Generation. 2024.

[2] Motamed, et al. Do Generative Video Models Understand Physical Principles? 2025.

[3] Li, et al. WorldModelBench: Judging Video Generation Models As World Models. 2025.

**Other Strengths And Weaknesses:**

Strengths:

- The idea of appearance-motion joint learning in video generation is very intuitive, as the model would be more effective at ensuring motion consistency.
- The comprehensive experimental analyses provide valuable insights into the original DiT and the new proposed model's behavior.
- The paper is well formatted and clearly written, making it easy to follow.

Weaknesses:

* **W3**: Reduced flexibility. Since the optical flow can only be computed within a single scene, the proposed framework cannot effectively handle scene cuts. In this sense, the appearance-motion joint learning framework sacrifices flexibility on multi-scene video clips for better single-scene consistency.
* **W4**: Inference overhead due to inner-guidance. According to Equation (8), each sampling step requires three evaluations, which imposes 1.5x inference cost compared to standard CFG. It would be better to discuss this and make comparisons under the same NFE budget rather than the same number of sampling steps.

**Questions For Authors:**

* **Q1**: What is the computational cost of extracting optical flow for 3M fine-tuning samples? Is it possible to use coarser but more efficient motion representations such as motion vector?

**Relation To Broader Scientific Literature:**

The paper advances video generation [Hong'22, Kondratyuk'23, Bar-Tal'24, Brooks'24] by learning an appearance-motion joint representation. Its improved physics capability contributes to world models [Ha'18, Brooks'24].

**Theoretical Claims:**

The paper makes no theoretical claims.

---

> ### Author Rebuttal · Authors · 2025-03-28
>
> We thank the reviewer for the comprehensive feedback and the useful points for discussion. Please find below our response.
>
> __VBench prompts:__ Both VBench and Movie Gen are common benchmarks for general video evaluation. The VBench prompts are not used in our work since out of the 11 dimensions for evaluation, _6 purposefully contain prompts with no motion at all_ (e.g., “object class” contains objects: “a person”, “a bicycle”, “temporal flickering” explicitly demands no motion, “a laptop, frozen in time”). Therefore, similar to recent leading models (Step-Video-T2V, Veo2), we employ Movie Gen, which evaluates all aspects of the generation (similar to VBench) while also incorporating motion (for more details, see our response to reviewer RHDu).
>
> __VideoJAM scalability:__ We appreciate the opportunity to clarify. While it is true that (only) the automatic metrics favor the 4B model, please note:
> 1. It is well-documented that automatic metrics often fail to align with human preferences [1,2,3]. For instance, Movie Gen dedicates a section (3.5.3) to metrics, highlighting that "across all tasks, we find that _existing automated metrics struggle to provide reliable results_”.
> Automatic metrics tend to prefer static motion since it is more “smooth” (as noted in the VBench paper, "a completely static video can score well in the aforementioned temporal quality dimensions"). Indeed, the 4B model scores better than the 30B model in the automated metrics, despite the 30B model being overwhelmingly superior (see human evaluation below). This anomaly can be attributed to the 4B model's tendency to produce videos with reduced motion (DiT-4B scores only 38.3 in its dynamic degree), which aligns with the biases of automated metrics discussed above.
> |**Model**|**Text Faith.**|**Quality**|**Motion**|
> |:------------:|:----------:|:-------:|:------:|
> |VideoJAM-4B|21.9%|21.1%|18.7%|
> |**VideoJAM-30B**|**78.1%**|**78.9%**|**81.3%**|
>
> [1] Polyak et al., Movie Gen.
> [2] Bartal et al., Lumiere.
> [3] Girdhar et al., Emu Video.
>
> 2. The [official VBench leaderboard](https://tinyurl.com/33tu3we5) shows similar inconsistencies. Models ranked highly by the [HuggingFace (HF) human leaderboard](https://tinyurl.com/28ea397z) are ranked significantly lower by VBench. For example, Kling is the leading baseline by both our human evaluation (Tab. 2) and HF. However, it is ranked 21st in the VBench leaderboard, below even much smaller models like CogVid-5B and Wan2.1-1.3B, both ranked much lower in the human leaderboard.
>
> 3. To further show robustness and scalability, we enclose a study in the same setting as Tab. 2, conducted on DiT-30B at a 512x512 resolution. VideoJAM maintains its significant improvement with a _71.1% lead in motion_, which is less than 2% below the improvement for the lower resolution.
> |**Model**|**Text Faith.**|**Quality**|**Motion**|
> |:------------:|:----------:|:-------:|:------:|
> |DiT-30B (512)|30.5%|26.6%|28.9%|
> | **+VideoJAM** |**69.5%**|**73.4%**|**71.1%**|
>
> __Additional references:__ Thank you for the useful suggestions, the related works section will be revised to include them.
>
> __Scene cuts:__ We hypothesize that it is still possible to use VideoJAM by applying RAFT on each scene separately and stitching the scenes with a “white” (no motion) frame.
>
> __Inner-Guidance latency:__ Inner-Guidance is applied to only 50% of the steps and performed in parallel using a batch size of 3 (L.726-731). This results in a 1.15x slowdown compared to standard CFG, which is a significantly lower slowdown than other multi-guidance methods (e.g., Instruct Pix2Pix incurs a 1.25x slowdown as it operates on all steps).
>
> __Physics benchmark:__ Thank you for this suggestion. We arbitrarily used the first benchmark suggested, and conducted an evaluation in the same setting as Tab. 2. The results below indicate a clear advantage to VideoJAM (62.7% preference in motion).
> Note that it is expected for the improvement in this benchmark to be somewhat decreased since optical flow is not an explicit physics representation, therefore it does not solve all physics incoherences (L. 412-418). As we mention in the conclusions, incorporating physics priors in future works is a fascinating direction. Samples from the comparison are provided in [this link](https://tinyurl.com/yhtmvx8p). We observe that these prompts are quite different from the ones employed by other benchmarks, and do not describe typical natural scenes.
> |**Model**|**Text Faith.**|**Quality**|**Motion**|
> |:------------:|:----------:|:-------:|:------:|
> |DiT-30B|48.2%|49.2%|37.3%|
> |+VideoJAM|**51.8%**|**50.8%**|**62.7%**|
>
> __Optical flow calculation on data:__ The calculation takes 13 hours in total (~1-2 seconds per sample with a batch size of 128).
> VideoJAM is a generic approach that can theoretically work with any representation, as long as there is a reliable way to encode it (e.g., RGB representation or a specialized VAE).
>
> We are happy to address any remaining questions.

---

> > ### Comment · Reviewer_VpZA · 2025-04-03
> >
> > Thank you for the detailed response, which addresses most of my concerns. I have one remaining question related to W1 (in line with with Reviewer RHDu):
> >
> > How does VideoJAM perform on VBench dimensions that are not directly related to motion? I can imagine a potential tradeoff between preserving static appearance details and modeling dynamic motion. Have you observed any other interesting findings?

---

> > > ### Author Response · Authors · 2025-04-04
> > >
> > > We thank the reviewer for the continued discussion. We are delighted that our previous response answered the concerns raised in the original review.
> > >
> > > Following the request by reviewer RHDu, please find below a table with the additional 10 evaluation axes of VBench for completeness of evaluation (the other 6 are already reported in the Appendix). The experiments were conducted using the official VBench prompts, in the setting described in the VBench paper, on the more powerful 30B variant of our model.
> > >
> > > We observe that in almost all axes (apart from the “scene” axis which describes static scenes such as “alley”, “amusement park”), the VideoJAM variant is either superior or equivalent to the base model. In axes that measure meaningful motion such as “human action”, VideoJAM is, as can be expected, superior to the base model. Note that as pointed out in our original response, some of the VBench temporal axes measure _frozen motion_ (e.g., “temporal flickering” is evaluated on purposefully static scenes). In these cases, we do not expect VideoJAM to improve on the performance of the base model.
> > >
> > > The results below are aligned with the intuition outlined in the paper, demonstrating that motion and appearance can be complementary. When the motion is coherent, it reduces body and object deformations and temporal artifacts, which in turn helps improve the perceived visual quality in videos (see SM website for many such examples). Additionally, we observe that VideoJAM does not harm the model’s ability to produce static scenes (see examples of such static videos from the VBench "frozen" evaluation prompts in [this link](https://tinyurl.com/r9y7h4jw)), which correspond to “white” (no motion) flow predictions.
> > >
> > > Following the reviews, we will add the table to the Appendix in the final version of the manuscript.
> > >
> > > | **Model** | **temporal flickering** | **object class** | **multiple objects** | **human action** | **color** | **spatial relationship** | **scene** | **appearance style** | **temporal style** | **overall consistency** |
> > > | :----------: | :--------------------: | :--------------: | :------------------: | :--------------: | :------: | :----------------------: | :-------: | :-------------------: | :------------------: | :-----------------------: |
> > > | DiT-30B      |        **99.66%**         |      89.07%       |        62.99%         |      97.00%       |  81.23%   |        72.01%          |   **52.07%**   |         22.05%         |         24.19%         |           27.53%           |
> > > | VideoJAM   |       **99.66%** |     **90.65%** |       **70.00%** |     **99.00%** | **91.09%** |       **73.32%** |  49.74% |        **23.25%** |        **24.42%** |          **27.57%** |

---

### Official Review · Reviewer_QjbW · 2025-03-13

**Overall Recommendation:** 4

**Summary:**

Despite recent advancements, generative video models still exhibit significant limitations in temporal coherence, especially when modeling real-world dynamic interactions and physics. The authors identify that this issue arises fundamentally from the traditional pixel-based reconstruction objectives, which prioritize appearance quality at the expense of coherent motion. To address this critical limitation, this paper proposes VideoJAM, a framework designed to explicitly integrate motion priors into video generation. Specifically, VideoJAM introduces a unified latent representation to jointly capture appearance and motion information by predicting both signals simultaneously through the addition of two linear layers during training. Additionally, during inference, the authors propose Inner-Guidance, a novel mechanism that dynamically adjusts the sampling process based on the model's own motion predictions. Extensive experiments demonstrate that VideoJAM substantially improves motion coherence, consistently outperforming several state-of-the-art open-source and proprietary models.

**Claims And Evidence:**

The authors' core method (VideoJAM) is evaluated comprehensively on the proposed VideoJAM-bench which is based on Movie Gen benchmark, utilizing clearly defined evaluation standards such as VBench metrics and the two-alternative forced choice (2AFC) human evaluation protocol. Experimental results convincingly demonstrate VideoJAM’s substantial improvement in motion coherence. However, the authors' claim regarding the adaptability and generality of VideoJAM to arbitrary video generation models currently lacks sufficient experimental validation. Specifically, all provided experiments exclusively employ DiT-based architectures (DiT-4B and DiT-30B), and no additional evidence is presented to demonstrate the generalization capability of VideoJAM to other prevalent architectures. Further validation using diverse, non-DiT video generation models is necessary to robustly substantiate this claim.

**Essential References Not Discussed:**

No

**Experimental Designs Or Analyses:**

The authors utilize not only the established Movie Gen benchmark but also introduce the VideoJAM-bench specifically designed to evaluate motion coherence, enhancing the relevance of their experimental setup. The selection of advanced baselines, including open-source models such as CogVideo5B and Mochi, and proprietary models such as Sora and Kling, ensures fairness and reliability of comparisons. The ablation studies are thorough and clearly demonstrate the contribution of text guidance, motion guidance, and the Inner-Guidance mechanism individually.
However, the experiments are limited by relatively low video resolutions and short durations (5 seconds), which may not fully reflect the model’s effectiveness in realistic, high-resolution, and longer-duration scenarios. Additionally, although a clear comparison standard (2AFC) was provided, the paper lacks detailed information on the selection criteria of human raters and the explicit evaluation protocols, potentially introducing subjectivity and bias into the experimental outcomes.

**Methods And Evaluation Criteria:**

The proposed VideoJAM framework is clearly motivated, conceptually straightforward, and innovative, effectively addressing the key issue of temporal coherence in generative video models. Specifically, the introduction of a joint appearance-motion representation and the Inner-Guidance mechanism provides explicit and effective guidance toward coherent motion predictions, which is theoretically sound and well justified. Additionally, the authors propose the VideoJAM-bench, specifically constructed based on the Movie Gen benchmark, focusing explicitly on challenging motion categories, including basic motion, complex motion, rotational motion, and physics-based interactions. The benchmark is well-designed, targeted, and capable of effectively evaluating motion coherence. Furthermore, the evaluation criteria, which include both automatic metrics (VBench) and human assessments (2AFC), are thorough and rigorous, comprehensively capturing the performance of the proposed approach.
However, despite the authors emphasizing the widespread use of the datasets without any modification, the relatively low resolution and short duration of the videos utilized may limit the ability to demonstrate the framework's practical effectiveness under high-resolution, longer-duration scenarios. Therefore, it would be beneficial to validate the method further on higher-resolution and longer-duration videos to fully establish its practical applicability and generalizability in real-world settings.

**Other Comments Or Suggestions:**

No

**Other Strengths And Weaknesses:**

Although VideoJAM demonstrates superior motion coherence compared to existing methods, it still struggles with fine-grained motion details, such as hand movements, in complex scenarios (e.g., "On a rainy rooftop, a pair of hip-hop dancers lock and pop in perfect sync, bringing energy and rhythm to the stage," and "A panda breakdancing in a neon-lit urban alley, graffiti art providing a colorful backdrop"). This limitation suggests room for future improvement in handling intricate motion coherence.

**Questions For Authors:**

No

**Relation To Broader Scientific Literature:**

The authors clearly identify the prevalent issue of motion incoherence in existing diffusion-based video generation models and thoroughly analyze its root cause—namely, traditional pixel-based reconstruction objectives being insensitive to temporal dynamics. Inspired by Composable Diffusion Models (Liu et al., 2022) and Classifier-Free Guidance (Ho & Salimans, 2022), they propose a novel conditional sampling mechanism termed Inner-Guidance. Unlike previous methods, VideoJAM explicitly emphasizes dependencies among conditioning signals and between conditioning signals and model weights, rather than assuming their independence. By dynamically leveraging motion predictions generated by the model itself during sampling, VideoJAM substantially enhances the coherence of generated motion. Furthermore, compared to recent motion representation approaches (e.g., Shi et al., 2024; Wang et al., 2024), which typically treat motion as an external conditioning input, VideoJAM explicitly integrates motion prediction into the training objective, systematically improving both motion generation quality and video coherence.

**Theoretical Claims:**

The theoretical analysis presented in VideoJAM primarily revolves around two aspects: (1) the theoretical justification and motivation behind the proposed Joint Appearance-Motion Representation, and (2) the theoretical derivation and rationale of the Inner-Guidance mechanism. Specifically, the authors clearly articulate the theoretical reasoning behind jointly modeling appearance and motion representations, addressing the fundamental limitations of traditional pixel-based objectives. Furthermore, they rigorously discuss the theoretical distinctions between their proposed Inner-Guidance and existing methods such as Classifier-Free Guidance and InstructPix2Pix guidance, emphasizing the critical dependence between motion prediction signals and model weights, a scenario not supported by existing assumptions of independence between conditions.
However, the selection of specific Inner-Guidance weights appears empirical, lacking theoretical justification or sensitivity analysis. A theoretical explanation or sensitivity analysis supporting the effectiveness of these particular weight values would further strengthen the theoretical rigor and clarity of the presented claims.

---

> ### Author Rebuttal · Authors · 2025-03-28
>
> We thank the reviewer for the comprehensive feedback and the interesting points for discussion.
> Please find below our response to the points raised in the review.
>
> __VideoJAM adaptability:__ We appreciate the feedback. While all concepts of our work can be easily generalized to any backbone, we acknowledge that this work focuses on the de-facto leading architecture, DiT. Following the review, the writing will be revised and refined to reflect this and avoid confusion.
>
> __Higher resolution and longer videos:__ The duration of our generated videos is limited by that of the base model we employ, which is 5 seconds (128 frames at 24 fps). However, to showcase the scalability of VideoJAM, we repeat the experiment from Tab. 2 on VideoJAM-30B with a higher resolution of 512x512. The results of the human study are enclosed below. As can be observed, VideoJAM is, once again, superior to the base model in all aspects, and very significantly improves motion coherence, further establishing the scalability of our method.
> | **Model** | **Text Faithfulness** | **Visual Quality** | **Motion** |
> |:------------:|:----------:|:-------:|:------:|
> | DiT-30B (512) |  30.5%    |  26.6%   |  28.9%   |
> | **+ VideoJAM** |  **69.5%** |   **73.4%** |   **71.1%** |
>
> __Inner-Guidance sensitivity test:__ Thank you for this suggestion. In accordance with common practice in diffusion papers, we selected the best scale empirically. Following the review, we enclose a qualitative sensitivity test with different Inner-Guidance scales in the [following link](https://tinyurl.com/3fx38map) (best viewed in full screen). All results are extracted in the same setting described in the paper.
> As can be observed, removing Inner-Guidance causes a noticeable degradation to the motion coherence (e.g., the legs of the man, the helicopter is flying backward). Importantly, Inner-Guidance demonstrates robust results across all reasonable scales (3,7) and does not display unusual sensitivities.
> When significantly increasing the scale (50), the results are out of distribution and cause a degradation in the video quality and motion, which is to be expected with any other guidance signal as well (e.g., text with CFG).
>
> __Human evaluation protocol:__ Thank you for raising this point, we appreciate the opportunity to clarify this aspect of our work. The human evaluators selected to participate in our study have extensive experience with evaluating generative visual models. As a filtering criterion, each evaluator has performed _at least 1000 evaluations_ before, where _at least 90% of those evaluations have been approved by third-party examiners_. The two videos in each comparison are randomly shuffled to ensure an unbiased comparison, and all videos are generated without watermarks to avoid identification.
> Additionally, note that the rating of the baselines by our evaluators (Tab. 2) is very much in correlation with the [public HuggingFace video leaderboard based on human evaluations](https://tinyurl.com/28ea397z), where Kling 1.5 is the leading baseline, with Sora being the second strongest baseline.
>
> We enclose below our instructions for the evaluators for transparency.
>
> > Hello! We need your help to read a caption, and then watch two generated videos. After watching the videos, we want you to answer a few questions about them:
> >Text alignment: Which video better matches the caption?
> >Quality: Aesthetically, which video is better?
> >Motion: Which video has more coherent and physically plausible motion? Do note, it is OK if the quality is less impressive as long as the motion looks better.
>
>
> We are happy to address any remaining questions.

---

### Official Review · Reviewer_kPFe · 2025-03-14

**Overall Recommendation:** 3

**Summary:**

This paper presents VideoJAM, a framework that improves motion coherence in generative video models by learning a joint appearance-motion representation. It introduces two key components: predicting both pixels and motion during training, and Inner-Guidance for coherent motion during inference. VideoJAM outperforms competitive models in motion coherence and visual quality, highlighting the importance of integrating appearance and motion for better video generation.

**Claims And Evidence:**

Claims: VideoJAM improves motion coherence in video generation models by learning joint appearance-motion representations, using dynamic guidance to steer generation toward coherent motion.

Evidence: The paper provides comparisons showing VideoJAM outperforms competitors in motion coherence and visual quality. However, the ablation study raises doubts about the necessity of the Inner-Guidance component, as it has little effect on results.

**Essential References Not Discussed:**

No

**Experimental Designs Or Analyses:**

The paper includes ablation studies to assess the impact of different components like Inner-Guidance and optical flow. However, the removal of Inner-Guidance shows minimal effect, and the correlation between Human Eval and Auto Metrics is unclear.

**Methods And Evaluation Criteria:**

Methods: VideoJAM introduces two key components: predicting both pixels and motion during training, and using Inner-Guidance during inference to enhance motion coherence.

Evaluation Criteria: The model’s performance is evaluated based on motion coherence, visual quality, and quantitative metrics (Human Eval, Auto Metrics), but the Human Eval results for the final method are not provided.

**Other Comments Or Suggestions:**

No

**Other Strengths And Weaknesses:**

Strength:
1.	The idea of using the model’s own evolving motion prediction as a dynamic guidance signal to steer generation toward coherent motion is interesting and highlights the difference from previous work.
2.	The presentation is clear and easy to understand.

Weakness:
1.	The introduction of motion representation can significantly improve the effectiveness of motion, and this conclusion is predictable. However, I noticed in the ablation study in Table 3 that removing Inner-Guidance has little effect on the results, while removing optical flow has the largest impact. This raises concerns about the necessity of Inner-Guidance. Additionally, I would like the authors to provide information on the impact of Inner-Guidance on inference time.
2.	I found that there is no correlation between the Human Eval and Auto. Metrics in Tables 1, 2, and 3, and the authors did not provide the Human Eval results for the final method. Therefore, I remain skeptical about the true effectiveness of the work.

**Questions For Authors:**

No

**Relation To Broader Scientific Literature:**

The paper builds on existing video generation models but emphasizes the novel integration of motion and appearance. However, it could engage more with related work on motion coherence and the role of dynamic guidance in generative models.

**Theoretical Claims:**

The paper suggests that integrating motion and appearance representations during training leads to more coherent motion in generated videos, which is a significant improvement over previous methods.

---

> ### Author Rebuttal · Authors · 2025-03-28
>
> We thank the reviewer for the comments and points for discussion. Please find below our response.
>
> __Human evaluation:__
> Thank you for your feedback. There appears to be a misunderstanding in the review. Importantly, _VideoJAM appears in all human evaluations_. As highlighted in all table captions, _the human evaluation scores indicate the percentage of votes that favor VideoJAM_, where each result uses the Two-Alternative Forced Choice (2AFC) protocol (Tabs. 1-3, L. 371-377). For example, a motion score of 85.9% for CogVid-5B indicates that evaluators prefer the motion by VideoJAM over CogVid-5B in 85.9% of votes. Since all rows are inherently comparative to VideoJAM, the final row ("VideoJAM vs. VideoJAM") contains minus signs (-). To further improve clarity, we will add visual graphs of the comparisons in the Appendix, and replace the minus signs with upper arrows.
> As noted in the paper (L. 400-403, L. 413-419), human raters consistently and significantly prefer VideoJAM in terms of motion over all baselines.
>
> __Impact of Inner-Guidance:__
> Inner-Guidance is a highly beneficial component of our method. As shown in Tab. 3, human evaluators significantly prefer the results with Inner-Guidance across all categories (68.9% vs. 31.1% for text faithfulness, 64.4% vs. 35.6% for quality, and 66.2% vs. 33.8% for motion). Also, note that _removing the optical flow prediction inherently removes Inner-Guidance_, as Inner-Guidance depends on the motion prediction. Thus, eliminating optical flow is an ablation that includes _both Inner-Guidance, and the flow prediction_ and, as expected, results in a greater performance drop.
> To further demonstrate the benefits of Inner-Guidance, we enclose a [qualitative ablation test](https://tinyurl.com/5c3jdab7) (best viewed in full screen). The results, generated in the same setting as in the paper, illustrate that removing Inner-Guidance often degrades motion coherence and introduces artifacts.
>
> __Inner-Guidance latency:__
> As noted in the implementation details (L.726-731), Inner-Guidance is applied to only 50% of the steps (following Sec. 3) and performed in parallel using a batch size of 3. This results in a 1.15x slowdown compared to the standard CFG, which is a significantly lower slowdown than other multi-guidance methods (e.g., Instruct Pix2Pix incurs a 1.25x slowdown since it operates on all steps).
>
> __Additional related works:__
> We appreciate the suggestion. The related works section will be revised to include all references proposed by the reviewers, ensuring a more comprehensive review of existing literature.
>
> __Automatic metrics vs. human evaluation:__
> Thank you for raising this point. We appreciate the opportunity to clarify.
> 1. It is well-documented in literature that automatic metrics often fail to align with human preferences [1,2,3]. For instance, the Movie Gen paper dedicates an entire section (3.5.3) to metrics, and highlights that "across all tasks, we find that _existing automated metrics struggle to provide reliable results_”.
> Specifically, temporal metrics tend to prefer either static motion since it is more “smooth” (as noted in the VBench paper "a completely static video can score well in the aforementioned temporal quality dimensions."), or completely incoherent videos that contain a lot of movement (Appendix, L. 767-768). Appearance metrics inherently prefer static videos, since they are typically based on _image models_, for which video frames that contain motion are out of distribution. All of the above underscores the inherent limitations of automated metrics in capturing the dynamic and nuanced nature of video content, and the necessity for a human-based evaluation.
> [1] Polyak et al., Movie Gen.
> [2] Bartal et al., Lumiere.
> [3] Girdhar et al., Emu Video.
>
> 2. The [official VBench leaderboard](https://tinyurl.com/33tu3we5) further highlights these inconsistencies. Models ranked highly by the [HuggingFace (HF) human leaderboard](https://tinyurl.com/28ea397z) are ranked significantly lower by VBench. For example, Kling 1.5 is the leading baseline by both our human evaluation (Tab. 2) and HF. However, it is ranked 21st in the VBench leaderboard, below even much smaller models like CogVid-5B and Wan2.1-1.3B, both ranked much lower in the HF human leaderboard.
>
> We enclose both automated metrics and human evaluations for completeness and believe continued research on reliable automatic metrics is essential for video generation assessment.
>
> __Supplementary materials (SM):__
> We noticed that the reviewer indicated “no” to viewing the SM. While we understand that reviewing SM is optional, we believe that the actual video results are very important to fully appreciate the improvement enabled by VideoJAM. We kindly ask the reviewer to consider viewing the results on our SM website.
>
> We are happy to address any remaining questions, and respectfully ask the reviewer to reconsider the score if our response has addressed all the questions.

---

### Official Review · Reviewer_RHDu · 2025-03-14

**Overall Recommendation:** 4

**Summary:**

The authors present Video-JAM (Joint Appearance-Motion representation), at an aim to capture real-world motion, dynamics, and physics, which existing video generative models struggle with handling.
In particular, the authors discover that the current video model training objective biases models towards fidelity, at the expense of motion coherence or temporal consistency.
To this end, the authors propose to train video generative models to predict the motion of the video - represented by optical flow masks - as well, thereby training the models on joint appearance (pixel) - motion (optical flow) representations.
This addition requires addition of a few layers (to handle the extra input and output), and training together **with** the optical flow supervision can only happen at the final finetuning stage.
Now that the video generative model is equipped with the ability to recover the motion, the authors propose the Inner Guidance to steer the model towards coherent motion.
VideoJAM achieves state-of-the-art performance in motion coherence, while also improving the visual quality of the generated videos.

**Claims And Evidence:**

Yes, the claims made in the submission are supported by clear and convincing evidence.

Claim 1: The current objective used for training video generative models is biased towards pixel fidelity, and neglects motion coherence.
- Substantiated by Figure 3 and Figure 8.

Claim 2. VIdeoJAM also improves visual quality of generated videos, not only the motion coherence.
- Validated in the quantitative/qualitative results in the experiment section.

**Essential References Not Discussed:**

While not an essential reference, adding Wonder3D as a reference may help the readers to understand that the idea of engaging additional modalities is a sound way of improving generative quality. Otherwise, I could not find any essential references which was not discussed.

**Experimental Designs Or Analyses:**

The experimental designs and analyses seem sound and valid. In particular, it was nice to see that human evaluation together with the automatic metrics from VBench, since the automatic metrics may not directly reflect the actual human preferences.

**Methods And Evaluation Criteria:**

Yes.
The authors use both human evaluation and the automatic metrics from VBench, the currently standard suite for evaluating video generative capabilities.

**Other Comments Or Suggestions:**

Please refer to W1 mentioned above.

**Other Strengths And Weaknesses:**

Strengths:

S1. Strong motivation and effective results.

S2. Simple and straightforward methodology without complications, yielding effective results.

Weaknesses:

W1. Not a strong weakness, but the paper does not include ALL the evaluation metrics provided in VBench. More results are included in Tables 4 and 5, but it would help to include the full results from VBench for the authors to fully understand the capabilities and shortcomings of VideoJAM.

**Questions For Authors:**

None

**Relation To Broader Scientific Literature:**

Similar endeavors were carried out in the text-to-3D / multi-view domain, where Wonder3D (Long et al., 2024) attempts to predict the normal map together with the pixel values during the generation process. Such work, including VideoJAM, hints that appropriate usage of additional modalities may help improve the generative results while not harming the generative quality.

**Theoretical Claims:**

Yes - section 4.3 sufficiently shows that since $d_t$ depends on the prompt and model weights (i.e., not independent with the model weights), the conventional CFG formulation cannot be used to guide motion coherence. The proposed inner guidance in Eq. 8 is derived with substantial motivation and correctness.

---

> ### Author Rebuttal · Authors · 2025-03-28
>
> We thank the reviewer for the comprehensive review of our work and the insightful suggestions. Please find below our response to the points raised in the review.
>
> __Related work on text-to-3D:__ Thank you for bringing this work to our attention. We will revise the related works section to reference this paper in the final version of the manuscript.
>
> __VBench metrics:__ Thank you for this point for discussion. VBench has two different contributions: a benchmark and a set of metrics. The benchmark is not used in our work since a significant part of the prompts there _intentionally describe static scenes_. Most categories of the benchmark (e.g., architecture, food, scenery, plants) are dominated by prompts that do not elicit any motion (e.g., “an apartment building with balcony”, “video of a indoor green plant” (grammar mistake in the original prompt), “landscape agriculture farm tractor”). Even the categories that entail motion (e.g., humans and animals) tend to focus on prompts that do not require meaningful temporal understanding (e.g., “people sitting on a sofa”, “a man with a muertos face painting”, "a black dog wearing halloween costume”).
> Additionally, out of the 11 dimensions for evaluation, _6 purposefully contain prompts with no motion at all_ (e.g., the class “object class” simply contains a list of objects such as “a person”, “a bicycle”, the class “color” contains prompts that assign colors to objects such as “a red bicycle”, “a green bicycle”). Therefore, similar to concurrent leading video models (Step-Video-T2V, Veo2) we opt to employ the Movie Gen benchmark as a general quality evaluator since it is carefully curated to evaluate all aspects of the generation (similar to VBench) while also explicitly incorporating motion instructions in all the prompts (see Section 3.5 of the [Movie Gen paper](https://arxiv.org/pdf/2410.13720) for details on the Movie Gen benchmark).
>
> The set of metrics proposed by VBench is divided into two: those that are applied by VBench to external benchmarks (see [official implementation](https://tinyurl.com/yc3xhyk5)) and a set that includes additional metrics that apply only to the VBench prompts. An example of such a metric is the VBench metric to estimate temporal flickering which requires _frozen videos_, thus the [prompts explicitly encourage no motion at all](https://tinyurl.com/yyd2z9b9). Naturally, we used _all metrics that are supported for external benchmarks_ to evaluate the performance of VideoJAM in the setting that it is intended for. This allows us to benefit from the generic and meaningful prompts from Movie Gen with the disentangled axes of evaluation from VBench.
>
> Notably, both benchmarks are not suitable for measuring motion coherence, as both tend to focus more on appearance and less on temporal understanding (L. 330-357). Thus, we construct an additional benchmark (VideoJAM-bench) to estimate temporal coherence and employ Movie Gen for a general assessment of the video quality.
>
> We are happy to address any remaining questions by the reviewer.

---

> > ### Comment · Reviewer_RHDu · 2025-04-03
> >
> > I thank the authors for their thoughtful answer to my comments.
> >
> > **VBench metrics:**  I appreciate the clarification provided, as well as your development of VideoJAM-bench to better assess temporal coherence. I recognize that your primary focus is on advancing motion generation, and I understand that many existing evaluation metrics, particularly those in VBench, do not adequately capture motion-related aspects.
> >
> > However, while motion generation is central to your contribution, video generation as a field encompasses more than just motion. Even though your work introduces the Movie Gen benchmark to evaluate human perceptual quality, many recent video generation methods include comparisons using the full VBench benchmark (including newer iterations such as VBench++ or VBench2.0). Reporting these additional metrics would provide a more comprehensive evaluation and allow the community to better understand the strengths and limitations of your method across all facets of video generation.
> >
> > I fully appreciate the novelty of your approach and the significant motion improvements demonstrated in the supplementary qualitative results. Including the full VBench results, even for categories not directly related to motion, would enhance the overall context of your work and help readers situate your contributions within the broader landscape of video generation research.

---

> > > ### Author Response · Authors · 2025-04-04
> > >
> > > We thank the reviewer for the continued discussion. We are delighted that the reviewer appreciates the novelty and the significance of our approach and we are grateful for the opportunity to further demonstrate the robustness of VideoJAM.
> > >
> > > Please find below a table with the additional 10 evaluation axes of VBench for completeness of evaluation (the other 6 are already reported in the Appendix). The experiments were conducted using the official VBench prompts, in the setting described in the VBench paper, on the more powerful 30B variant of our model.
> > >
> > > We observe that in almost all axes (apart from the “scene” axis which describes static scenes such as “alley”, “amusement park”), the VideoJAM variant is either superior or equivalent to the base model. In axes that measure meaningful motion such as “human action”, VideoJAM is, as can be expected, superior to the base model. Note that as pointed out in our original response, _some of the VBench temporal axes measure frozen motion_ (e.g., “temporal flickering” is evaluated on purposefully static scenes). In these cases, we do not expect VideoJAM to improve on the performance of the base model.
> > > These results further substantiate the robustness of our method and its ability to improve motion coherence while maintaining, and even improving, the visual quality of the model. Following the reviews, we will add this table to the Appendix in the final version of the manuscript.
> > > | **Model** | **temporal flickering** | **object class** | **multiple objects** | **human action** | **color** | **spatial relationship** | **scene** | **appearance style** | **temporal style** | **overall consistency** |
> > > | :----------: | :--------------------: | :--------------: | :------------------: | :--------------: | :------: | :----------------------: | :-------: | :-------------------: | :------------------: | :-----------------------: |
> > > | DiT-30B      |        **99.66%**         |      89.07%       |        62.99%         |      97.00%       |  81.23%   |        72.01%          |   **52.07%**   |         22.05%         |         24.19%         |           27.53%           |
> > > | VideoJAM   |       **99.66%** |     **90.65%** |       **70.00%** |     **99.00%** | **91.09%** |       **73.32%** |  49.74% |        **23.25%** |        **24.42%** |          **27.57%** |

---

### Decision · Program_Chairs · 2025-05-01

**Decision:**

Accept (oral)

**Comment:**

This paper introduces VideoJAM, a straightforward approach for improving motion coherence and temporal consistency in video generation models. It works by denoising optical flow maps alongside the pixels, resulting in a joint appearance-motion representation. It is also shown how, during inference, the resulting approach can be steered towards generating videos with better motion using guidance.

3/4 of the reviewers acknowledged the author response and engaged with the authors during the rebuttal stage. Among those reviewers there is broad agreement that the contributions in this work are significant, timely, and that the claims are well supported by evidence. A few concerns, such has how VideoJAM affects performance on non-motion related categories, or whether this approach is scalable have been convincingly addressed by the authors as evidenced by all reviewers recommending accept. Remaining limitations, such as the method struggles more often with fine-grained motion categories are suitable for follow-up work. The AC agrees with the recommendation to accept and concurs that the work has the potential to be impactful and of broader interest to the video generative modeling community.